# TVBench: Redesigning Video-Language Evaluation

## Abstract

Large language models have demonstrated impressive performance when integrated with vision models even enabling video understanding. However, evaluating these video models presents its own unique challenges, for which several benchmarks have been proposed. In this paper, we show that the currently most used video-language benchmarks can be solved without requiring much temporal reasoning. We identified three main issues in existing datasets: (i) static information from single frames is often sufficient to solve the tasks (ii) the text of the questions and candidate answers is overly informative, allowing models to answer correctly without relying on any visual input (iii) world knowledge alone can answer many of the questions, making the benchmarks a test of knowledge replication rather than visual reasoning. In addition, we found that open-ended question-answering benchmarks for video understanding suffer from similar issues while the automatic evaluation process with LLMs is unreliable, making it an unsuitable alternative. As a solution, we propose TVBench, a novel open-source video multiple-choice question-answering benchmark, and demonstrate through extensive evaluations that it requires a high level of temporal understanding. Surprisingly, we find that most recent state-of-the-art video-language models perform similarly to random performance on TVBench, with only a few models such as Qwen2-VL, and Tarsier clearly surpassing this baseline.

## 1 Introduction

Vision Language Models have gained popularity, benefiting from both the progress made in natural language processing and the surge of foundation models for vision tasks with strong generalization capabilities. Recently, video-language models have been introduced (Xu et al., 2021; Lin et al., 2023; Xue et al., 2023), aiming to replicate the success achieved in the image domain. To evaluate their performance, visual question answering has emerged as a key task requiring both textual and visual reasoning. With the rapid model development and release cycles, having a reliable and robust benchmark is crucial in measuring progress and guiding research efforts.

There are two main approaches to designing question-answering benchmarks for videos: multiple-choice question answering (MCQA) (Wang et al., 2024c; Li et al., 2024b) and open-ended question answering (Yu et al., 2019; Xu et al., 2017). The most popular and commonly used benchmark for MCQA is MVBench (Li et al., 2024b), which is the most downloaded video dataset with over 104K monthly downloads as of September 2024 and the second most downloaded across both image and video visual question-answering benchmarks on Hugging Face despite being released only recently. As such, its reliability as a valid benchmark is of utmost importance. But how much *video* understanding does MVBench truly measure?

Previous analysis in image question answering benchmarks (Goyal et al., 2017b) has demonstrated that poorly formulated benchmarks could bias the development of new models towards learning strong text representations while ignoring visual information. This is especially relevant for the video-language community, where benchmarks must account not only for visual but also for temporal understanding.

In this work, we conduct a comprehensive analysis of widely used video question-answering benchmarks, revealing that temporal information is poorly evaluated. Furthermore, in MCQA tasks, prior world knowledge, combined with overly informative questions and answer choices, often allows questions to be answered solely through text, without the need for visual input. Our results also indicate that automatic open-ended evaluation is unreliable, with significant evaluation discrepancies in results across different models for the same task. We assess the substantial shortcomings of MVBench and propose as a solution a new benchmark TVBench that requires temporal understanding to be solved, providing an effective evaluation tool for current video-language models:

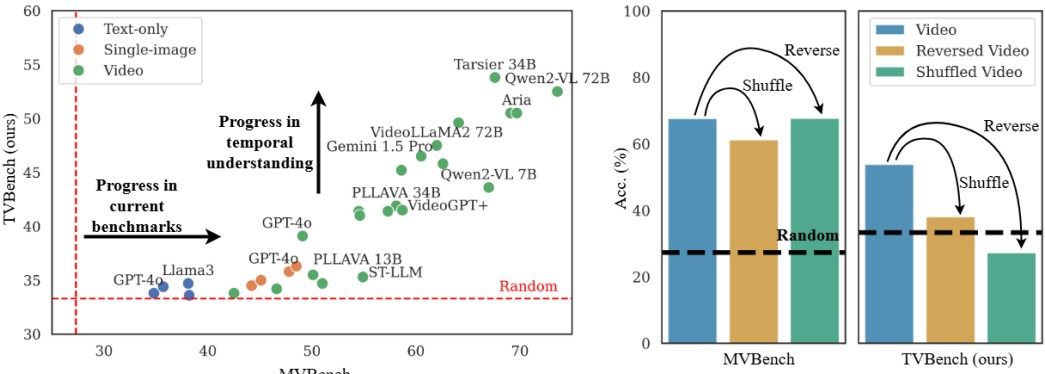

Figure 1: **TVBench a temporal video-language benchmark.** In TVBench, state-of-the-art text-only, image-based, and most video-language models perform close to random chance, with only the latest strong temporal models, such as Tarsier, outperforming the random baseline. In contrast to MVBench, the performance of these temporal models significantly drops when videos are reversed.

- MVBench contains unlikely candidate answers easily dismissed from a single frame.
  → We provide only temporal challenging candidate answers, requiring models to leverage temporal information to answer correctly.
- MVBench contains QA pairs with obvious solutions due LLM biased generation.
  → We design task-specific templates to generate questions that are not overly informative such that they cannot be answered solely by text.
- MVBench contains QA pairs that can be solved by solely relying on prior world knowledge.
  → We design questions that can only be answered from the video content, without relying on prior world knowledge.

As a result, TVBench measures the temporal understanding of video-language models in contrast to MVBench, as shown in Fig. 1. In this setting, text-only and single-frame models, such as Gemini 1.5 Pro and GPT-4o, perform at random chance levels on TVBench, despite achieving competitive results on MVBench. Surprisingly, even recent state-of-the-art video-language models perform close to random chance on TVBench, with only a few models such as Qwen2-VL and Tarsier outperforming the random baseline. For these models, shuffling and reversing the videos lead to significant performance drops, unlike in MVBench, further verifying TVBench as a temporal video benchmark.

## 2 RELATED WORK

Traditional video evaluation benchmarks focused on specific tasks such as action recognition (Goyal et al., 2017a; Kay et al., 2017) or video description (Das et al., 2013; Xu et al., 2016; Wang et al., 2019). With the emergence of Vision Language Models (VLMs), there is a growing need for more comprehensive evaluation protocols to effectively evaluate models with increasingly advanced generalization capabilities. Current video-language benchmarks focus on solving QA pairs that require a certain level of multimodal understanding. There are two major trends in the QA format: open-ended QA and multiple-choice QA (MCQA).

**Open-ended question answering.** Evaluating open-ended QA introduces new challenges, as traditional evaluation metrics such as ROUGE (Lin, 2004), METEOR (Banerjee & Lavie, 2005), and CIDEr (Vedantam et al., 2015) fail to analyze discrepancies of more complex and elaborated answers. Alternatively, Maaz et al. (2023) introduces a novel quantitative evaluation pipeline for open-ended QA datasets. The proposed method relies on GPT-3.5 to determine the correctness of the predicted answer and provides a matching score with the ground truth. Commonly used datasets for evaluating models in this context include MSRVTT-QA (Xu et al., 2017), MSVD-QA (Xu et al., 2017), TGIF-QA (Jang et al., 2017) and ActivityNet-QA (Yu et al., 2019). In general, any open-ended QA benchmarks can be evaluated following this protocol. As shown in our analysis, Large Language Model (LLM) based evaluations are prone to hallucinations, leading to unreliable conclusions. In contrast, MCQA benefits from a more straightforward evaluation process, based on the accuracy score.

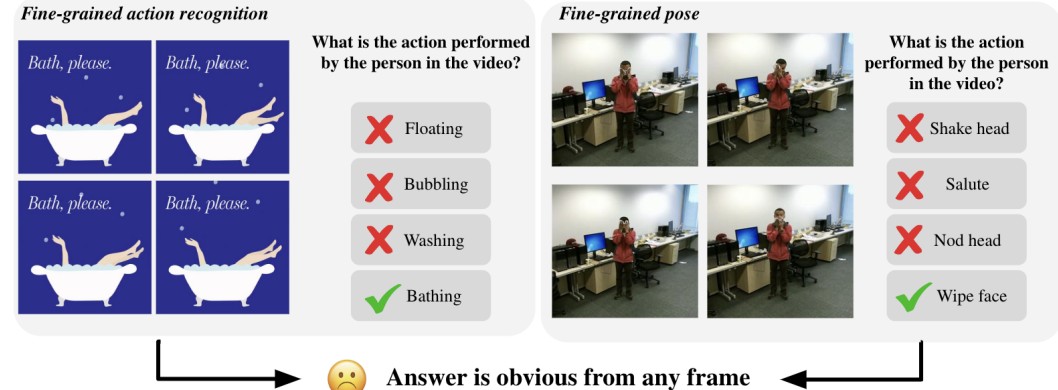

Figure 2: **Spatial bias of MVBench video-language benchmark.** We show different tasks of the MVBench benchmark and observe that the question can be answered without requiring temporal understanding. For quantitative results see Table 1.

**Multiple-choice question answering.** CLEVRER (Yi et al., 2019) assesses reasoning about object interaction in synthetic videos. Perception Test (Patraucean et al., 2024) was introduced to evaluate visual perception in multimodal settings, mainly in indoor scenes. (Bagad et al., 2023) uses synthetic generated data to evaluate the temporal understanding of early video-language models. EgoSchema (Mangalam et al., 2024) focuses on MCQA involving long egocentric videos. Recently, VideoHallucer (Wang et al., 2024c) was introduced as a first attempt to define a video-language benchmark specifically designed for hallucination detection. MVBench (Li et al., 2024b) aims to evaluate temporal understanding by defining 20 dynamic tasks specifically designed to require temporal reasoning throughout the entire video. However, our experiments demonstrate that many of these tasks are highly spatial and textual biased, failing to evaluate temporal understanding effectively. We propose a new benchmark that requires a high level of spatiotemporal understanding across different tasks in order to be solved.

## 3 PROBLEMS IN VIDEO MCQA BENCHMARKS

In this section, we identify two key shortcomings in current video multiple-choice question-answering benchmarks, as demonstrated on MVBench. First, we show that this benchmark contains strong spatial bias, meaning that questions can be answered without temporal understanding. Secondly, we demonstrate that MVBench also contains a strong textual bias, as many questions can be answered without even looking at the visual input. In Appendix A.2.1, we extend our analysis to NextQA (Xiao et al., 2021).

### 3.1 DOES TIME MATTER?

Video benchmarks must define tasks that cannot be solved using solely spatial information to effectively evaluate the temporal understanding of a model. In video MCQA, this means questions should not be answerable using spatial details from a single random frame or multiple frames e.g. after shuffling them. However, if no understanding of the sequence of events and temporal localization is needed, the benchmark fails to assess temporal understanding, focusing only on spatial information which we define as spatial bias.

We analyze the spatial bias in MVBench using state-of-the-art image and video-language models such as GPT-4o (OpenAI, 2024), Gemini 1.5 Pro (Gemini, 2024), and Tarsier-34B (Wang et al., 2024a). In Table 1 we focus on four different tasks i) scene transition ii) fine-grained pose iii) fine-grained action and iv) episodic reasoning. To assess whether solving these tasks requires temporal understanding, we compare the performance of such models when receiving only a single random frame, the shuffled videos, and the original videos.

The models receiving only a random frame as input show strong performance across all four tasks in Table 1, surpassing the random baseline. Notably, GPT-4o achieves the highest average performance of 62.8% across the four tasks, nearly matching its video performance with 65.8% and other state-of-the-art video-language models. The lower image performance of Tarsier-34B might stem from its training data composition, which contains five times more video data than image data. These findings are unexpected, as task names like *Fine-grained Action* suggest a need for temporal under-

Table 1: **Spatial bias of the MVBench video-language benchmark.** Our analysis reveals that temporal understanding is not required for solving these temporal tasks of MVBench as they can be solved with a random frame or shuffled video. Average represents the mean across these four tasks.

| | Input | Fine-grained Action | Scene Transition | Fine-grained Pose | Episodic Reasoning | Average |
|---|---|---|---|---|---|---|
| Random | – | 25.0 | 25.0 | 25.0 | 20.0 | 23.8 |
| Gemini 1.5 Pro | | 47.0 | 78.0 | 46.5 | 56.5 | 57.0 |
| GPT-4o | image | 49.0 | 84.0 | 53.0 | 65.0 | 62.8 |
| Tarsier-34B | | 48.5 | 67.0 | 22.5 | 46.0 | 46.0 |
| Gemini 1.5 Pro | | 50.0 | 93.3 | 58.5 | 66.8 | 67.2 |
| GPT-4o | video | 51.0 | 83.5 | 65.5 | 63.0 | 65.8 |
| Tarsier-34B | | 48.5 | 89.5 | 64.5 | 54.5 | 64.3 |
| Gemini 1.5 Pro | | 49.5 | 90.0 | 54.5 | 63.0 | 64.3 |
| GPT-4o | shuffle | 52.0 | 84.5 | 69.0 | 64.5 | 67.5 |
| Tarsier-34B | | 51.0 | 89.0 | 56.5 | 51.5 | 62.0 |

standing. For this fine-grained task, the image-model GPT-4o achieves 49%, which is even slightly better than the state-of-the-art model Tarsier, which scores 48.5%. Similarly, for the other three tasks. Overall, GPT-4o achieves an average accuracy across all 20 tasks of 47.8%, which is 20.5% higher than the random performance of 27.3% on MVBench. This indicates that a large portion of the benchmark is affected by spatial bias.

Additionally, shuffling the videos has minimal impact on the performance of all video-language models with an average difference of 2.3%, indicating that temporal information is not necessary to solve these tasks. Note, as confirmed in Sec. 5.3, the Tarsier model shows a significant drop in performance when videos are shuffled for tasks that require temporal understanding. This problem goes beyond these four tasks as shown in Table 4, Gemini 1.5 Pro and Tarsier achieve an average accuracy across all 20 MVBench tasks of 60.5% and 67.6%, respectively. Shuffling video frames causes a performance drop of only 3.8% and 6.4%, respectively, indicating that the spatial bias affects not only the tasks analyzed in Table 1 but the entire dataset. Additionally, we verify the agreement between the correct responses of Tarsier-34B across modalities: 91.0% between image and video inputs, and 93.9% between video and shuffled video. This confirms that current models heavily rely on spatial biases to solve MVBench.

In Fig. 2 we show examples corresponding to tasks analyzed in Table 1. The example from the left is taken from the *Fine-grained Action recognition* task implying that temporal understanding is needed. The example on the right is from the *Fine-grained Pose* task. In both cases, the information provided from any frame is enough to correctly answer the question. In Appendix, in Fig. 14 -20 we show 34 more examples of spatial bias in MVBench.

> **Problem 1**
>
> The MVBench benchmark has a strong spatial bias meaning questions can be answered without requiring temporal understanding.

## 3.2 DOES VISION MATTER?

Video benchmarks must be designed to prevent questions from being answered solely through common sense reasoning. Modern LLMs possess strong reasoning skills, which can exploit the information within the question and candidate sets in MCQA video language evaluation benchmarks. This creates textual bias, enabling models to answer questions without leveraging the video content.

We analyze the impact of textual bias on MVBench in Table 2. We evaluate the performance of state-of-the-art text-only LLMs, Llama 3 (MetaAI, 2024), and multi-modal LLMs such as Gemini 1.5 Pro (Gemini, 2024), GPT-4o (OpenAI, 2024) and Tarsier (Wang et al., 2024a). Our findings reveal that LLMs can eliminate incompatible candidates easily, greatly outperforming the random baseline. Models using only text achieve competitive results compared to video-language models across these four tasks (Action Count, Unexpected Action, Action Antonym, and Episodic Reasoning). For instance, Gemini 1.5 Pro achieves an average performance of 62.3% using text-only, compared to Tarsier-34B's 67.4% using videos. Additionally, we verify an 85.3% agreement between Tarsier-34B's correct text and video responses, confirming its strong reliance on textual biases in MVBench.

Figure 3: **Textual bias of MVBench video-language benchmark.** We show different tasks of MVBench and find that questions can be answered without taking the visual part into account.

This goes beyond the four tasks, as Gemini Pro 1.5 achieves an average performance across all 20 tasks of 38.2%, which is 10.9% higher than the random chance baseline of 27.3%. We have identified three key sources of this textual bias.

**Bias from LLM-based QA generation.** Collecting and manually annotating large datasets for training and evaluation is very costly. Automatic and semi-automatic collection and annotation processes are commonly used (Li et al., 2024b; Mangalam et al., 2024). This includes techniques such as automatic QA pair generation with LLMs. ChatGPT plays a fundamental role in QA generation for 11 of the 20 tasks in the MVBench dataset. However, this introduces unrealistic candidates and QA pairs with excessive information. Fig. 3 presents examples of QA pairs that can be resolved merely with text information. Questions 1 and 2 belong to the *Action Antonym* task, where an LLM is prompted to generate the antonym of the actual action shown in the video. The answers generated are either unrealistic, as one cannot "remove something into something," or consistently incorrect, such as "not sure." Questions 3 and 4 are from the *Unexpected Action* task. In this task, the LLM is prompted to generate textual questions and candidates based on the dataset annotations. However, in Question 3, the subject inquired in the question only occurs in the correct answer, while in Question 4, one of the candidates is a paraphrased version of the question. Hence, the text-only model is able to identify the correct answer without visual information.

**Bias from unbalanced sets.** Unbalanced QA sets also hinder a robust evaluation process. For instance, the correct answer for the Action Count task on MVBench is '3' for 90 out of 200 questions, while '9' is only the correct answer for one question. A model with a similar bias might get higher results than random by chance. We have observed in our experiments that some text-only models such as GPT-4o have this bias, predicting '3' for 88 out of 200 samples. This makes GPT-4o perform on par with the best video model with an accuracy of 44.0% and 46.5% respectively.

**Overreliance on world knowledge in questions.** Video benchmarks should ensure models cannot rely solely on memorized world knowledge from an LLM to guess answers without using visual input. Even with well-designed questions, models might bypass visual reasoning and rely on prior knowledge to answer correctly. An example of this can be seen in question 5 of Fig. 3. The question does not exhibit an obvious bias in the QA generation, but it can be correctly answered if the model has world knowledge of the TV show from which the question was derived as answer 2 are character names from the House TV show.

In Appendix, in Fig. 21 -25 we show 26 more examples of textual bias in MVBench.

> **Problem 2**
>
> The MVBench benchmark can be partially solved without visual information due to the bias from LLM QA generation, unbalanced dataset, and overreliance on world knowledge.

Table 2: **Textual bias of the MVBench video-language benchmark**. Our analysis reveals that vision is not required for solving tasks from MVBench as text-only LLMs score high above the random baseline and nearly on par with video models. Average is with respect to these four tasks.

| | Input | Action Count | Unexpected Action | Action Antonym | Episodic Reasoning | Average |
|---|---|---|---|---|---|---|
| Random | – | 33.3 | 25.0 | 33.3 | 20.0 | 27.9 |
| Llama3 70B | | 44.5 | 63.5 | 74.5 | 50.5 | 58.3 |
| Gemini 1.5 Pro | text- | 49.0 | 68.0 | 85.5 | 49.0 | 62.3 |
| GPT-4o | only | 44.0 | 69.5 | 57.5 | 51.5 | 55.6 |
| Tarsier-34B | | 37.0 | 39.5 | 66.0 | 44.0 | 46.6 |
| Gemini 1.5 Pro | | 41.2 | 82.4 | 64.5 | 66.8 | 63.7 |
| GPT-4o | video | 43.5 | 75.5 | 72.5 | 63.0 | 63.6 |
| Tarsier-34B | | 46.5 | 72.0 | 97.0 | 54.5 | 67.4 |

## 4 OPEN-ENDED QA TO THE RESCUE?

Contrary to multiple-choice question answering (MCQA), open-ended question answering can be seen as an alternative to solving the aforementioned issues. Without a predefined candidate answer set, the model cannot rely on textual information to eliminate implausible candidates. However, open-ended evaluation presents new challenges compared to MCQA. Following Maaz et al. (2023), LLMs have been widely used for the evaluation of open-ended question-answering in datasets such as MSVD-QA (Wu et al., 2017), MSRVTT-QA (Xu et al., 2016) and ActivityNet QA (Yu et al., 2019). Specifically, Maaz et al. (2023) proposed GPT-3.5 as the evaluator model, which makes the entire evaluation process rely on a private API model. The evaluation model is prompted to determine if the predicted answer is correct given the question and the ground-truth answer. In addition, the evaluator also computes a score to measure the answer quality.

We conducted a comparative analysis to assess the influence of the evaluation model on the results. Table 3 shows the accuracy and average score for different models on three open-ended datasets, using two evaluators: GPT-3.5 and Llama3-70B. The evaluators produced significantly different results for the same method on the same dataset, with discrepancies of more than 20 points. Specifically, Llama3 highly increases the accuracy of text-only and single-image models, while providing similar or even lower results than GPT-3.5 for video models. It is also evident that Llama3 assigns better metrics to predictions made by the same model. If both the prediction model and the evaluator contain similar biases, the hallucinations in the predictions may be classified as correct responses by the evaluator ignoring the correct answer. These findings raise doubt about the reliability of these evaluations, as different models give completely different results.

Moreover, as shown in Table 3 open-ended QA does not solve the main issues of MCQA. The performance of text-only models is surprisingly strong; state-of-the-art LLMs can guess the answer solely from the question text for a significant number of questions, even without a candidate list. This includes questions such as *Which hand of the person in black wears a watch?* or *What color is the pants of a person wearing black clothes?*, which correct answers are *Left hand* and *Black*. The first question can be answered just with prior knowledge as people commonly wear the watch on the left hand, while in the second one, the question gives too much information, the person is wearing black clothes. Similar to the findings for MCQA on spatial bias in Sec. 3.1, when using a single random frame for image-text models such as GPT-4o, performance reaches 60.6% and 46.4%, approaching the video-language model's 80.3% and 61.6%, respectively. In addition, the performance of Tarsier-34B does not significantly drop—on average less than 3%—when the input videos are shuffled, indicating the low temporal understanding required for solving the benchmarks. These results show that open-ended video-language benchmarks also exhibit strong spatial bias, not requiring temporal understanding to be solved.

Fig. 4 shows incorrect evaluations on ActivityNet QA and MSVD QA. In these experiments, we prompt a Llama3 model to answer the questions in a text-only setting and perform the evaluation with GPT-3.5. In the first question, it can be seen how the model provides an answer completely unrelated to the video: "Cacti, succulents, ocotillo, mesquite, creosote bush, ...," offering general information about desert plants. This is expected as the model only receives the question without visual information. The evaluation model classifies this question as correct with the highest matching score. It seems that the correct answer is disregarded in the evaluation of this example, focusing on

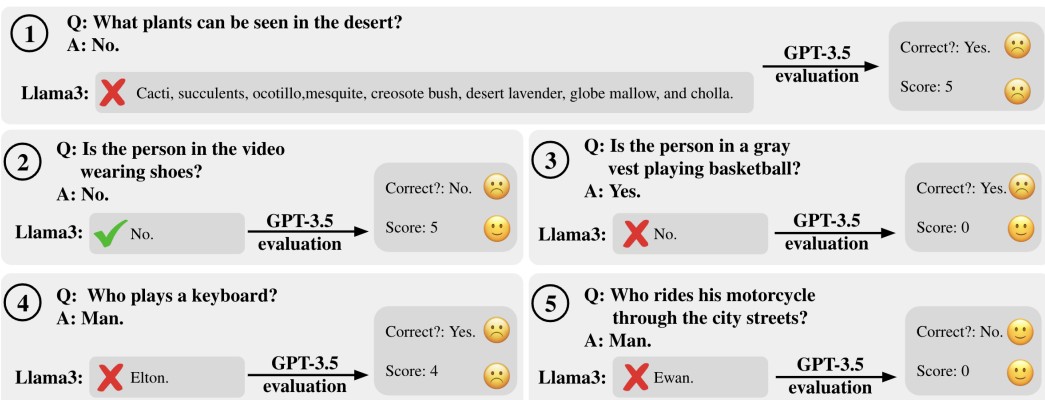

Figure 4: **Unreliability of open-ended video-language benchmarks.** GPT 3.5 is commonly used as an evaluator of open-ended responses, here we use Llama 3 in a text-only setting to generate answers. GPT gives confusing accuracies and scores. Smiley emoji shows truthful or unreliable evaluation from GPT 3.5.

the relationship between the questions and the predicted answer. Questions 2 and 3 contain two instances of no/yes QA pairs where the model correctly identifies whether they are correct, but assigns completely contradictory scores. Questions 4 and 5 contain evaluations in which the correct answer is *man*. However, it can be seen how the text-only model answers with specific names rather than a more general concept. The evaluations are completely different, most probably due to a context bias, as both Llama3 and the evaluation model –GPT3.5– links the concept *keyboard* with *Elton*.

In summary, current open-ended benchmarks are unreliable due to their use of LLMs as evaluators. This makes them unsuited for evaluating video-language models, especially as they also suffer from spatial and textual bias. In addition, they rely on closed-source LLMs for evaluation, which incurs costs to access, and becomes unreproducible when newer versions are released.

## 5 TVBENCH: A TEMPORAL VIDEO QUESTION ANSWERING BENCHMARK

We propose TVBench, a new benchmark specifically created to evaluate temporal understanding in video QA. We adopt a multiple-choice QA approach to prevent the problems of open-ended VQA described in Sec. 4. The main design principles of TVBench are derived from, and address, the problems listed in Sec. 3. Appendix A.2 provides an overview of the tasks, questions, and answers candidates used in our benchmark. We verify our choice of tasks and QA templates in Sec. 5.3 by the performance of multi-modal LLMs with solely a random frame or shuffled videos.

### 5.1 DEVELOPING A TEMPORAL VIDEO-LANGUAGE BENCHMARK

This section explains the key strategies implemented in TVBench to address the issues identified in Sec. 3 of current video MCQA evaluation benchmarks.

**Strategy 1: Define Temporally Hard Answer Candidates.** To address Problem 1, it is crucial that the temporal constraints in the question are essential for determining the correct answer. This involves designing time-sensitive questions and selecting temporal challenging answer candidates.

1. We select 10 temporally challenging tasks that require: repetition counting (Action Count), properties of moving objects (Object Shuffle, Object Count, Moving Direction), temporal localization (Action Localization, Unexpected Action), sequential ordering (Action Sequence, Scene Transition, Egocentric Sequence), and distinguishing between similar actions (Action Antonyms).

2. We define hard-answer candidates based on the original annotations to ensure realism and relevance, rather than relying on LLM-generated candidates that are often random and easily disregarded, as seen in MVBench. For example, in the Scene Transition task (Fig. 5), we design a QA template that provides candidates based on the two scenes occurring in the videos for this task, rather than implausible options like "From work to the gym." Similarly, for the Action Sequence task, we include only two answer candidates corresponding to the actions that actually occurred in the video. More details for the remaining tasks in Appendix A.2.

Table 3: **Unreliability of open-ended video-language benchmark evaluation.** Different LLMs used for evaluation produce varying accuracies and scores, see $\Delta$ column. Additionally, open-ended benchmarks also exhibit spatial and textual bias, similar to MCQA.

| Benchmark | Model | Input | Evaluation method | | | | |
|---|---|---|---|---|---|---|---|
| | | | GPT-3.5 | | Llama3-70B | | $\Delta$Acc. |
| | | | Acc. | Score | Acc. | Score | |
| **MSVD-QA** | Llama3 70B | text | 29.1 | 2.6 | 51.9 | 2.8 | +22.8 |
| | GPT-4o | text | 29.0 | 2.5 | 44.2 | 2.4 | +15.2 |
| | GPT-4o | image | 60.6 | 3.6 | 75.3 | 3.8 | +14.7 |
| | Tarsier-34B | video shuffle | 76.7 | 4.0 | 78.3 | 4.1 | +1.6 |
| | Tarsier-34B | video | 80.3 | 4.2 | 78.3 | 4.1 | -2.0 |
| **MSRVTT-QA** | Lama3 70B | text | 23.9 | 2.4 | 47.8 | 2.6 | +23.9 |
| | GPT-4o | text | 23.3 | 2.3 | 42.4 | 2.3 | +19.1 |
| | GPT-4o | image | 34.2 | 2.7 | 50.8 | 2.7 | +16.6 |
| | Tarsier-34B | video shuffle | 63.1 | 3.5 | 62.7 | 3.4 | -0.4 |
| | Tarsier-34B | video | 66.4 | 3.7 | 63.0 | 3.4 | -3.4 |
| **ActivityNet-QA** | Lama3 70B | text | 25.3 | 2.6 | 32.6 | 1.9 | +7.3 |
| | GPT-4o | text | 27.1 | 2.5 | 33.9 | 1.8 | +6.8 |
| | GPT-4o | image | 46.4 | 3.2 | 56.2 | 2.9 | +9.8 |
| | Tarsier-34B | video shuffle | 59.9 | 3.6 | 60.8 | 3.4 | +0.9 |
| | Tarsier-34B | video | 61.6 | 3.7 | 61.3 | 3.4 | -0.3 |
| Standard deviation between GPT3.5 and Llama3 score differences: $\pm$ 9.3 | | | | | | | |

**Strategy 2: Define QA pairs that are not overly informative.** Contrary to LLM-based generation, we apply basic templates to mitigate the effect of text-biased QA pairs, mitigating Problem 2.

1. We design QA pairs that are concise and not unnecessarily informative by applying task-specific templates. These templates ensure that the QA pairs lack sufficient information to determine the correct answer purely from text. An example of Unexpected Action is illustrated in Fig. 6. QA pairs require the same level of understanding for the model to identify what is amusing in the video, but without providing additional textual information. Unlike MVBench, the model cannot simply select the only plausible option containing a dog. We use the same candidate sets across tasks like Action Count, Object Count, Object Shuffle, Action Localization, Unexpected Action, and Moving Direction, to ensure balanced datasets with an equal distribution of correct answers, keeping visual complexity while reducing textual bias. Appendix Table 5 provides an overview of all tasks, demonstrating that the QA templates are carefully crafted without unnecessary textual information.

2. Solving the overreliance on world knowledge requires providing questions and candidates that contain only the necessary information, specifically removing factual information that the LLM can exploit. We remove tasks such as Episodic Reasoning, that are based on QA pairs about TV shows or movies.

## 5.2 DATASET SOURCE

Videos in TVBench are sourced from Perception Test (Patraucean et al., 2024), CLEVRER (Yi et al., 2019), STAR (Wu et al., 2024), MoVQA (Zhang et al., 2023), Charades-STA (Gao et al., 2017), NTU RGB+D (Liu et al., 2019), FunQA (Xie et al., 2023) and CSV (Qian et al., 2022). Overall, TVBench comprises first and third-person perspectives, indoor and outdoor scenes, and real and synthetic data with 2,654 QA pairs among 10 different tasks. QA pairs are generated based on original annotations following the model provided in Appendix A.2 for each task.

## 5.3 TVBENCH EVALUATION

Table 4 provides a detailed performance breakdown of state-of-the-art text (MetaAI, 2024) and multi-modal LLMs (Zhang et al., 2024a; Li et al., 2024a; Zhang et al., 2024b; OpenAI, 2024; Maaz et al., 2024; Li et al., 2024b; Xu et al., 2024; Gemini, 2024; Wang et al., 2024a; Lin et al., 2023; Cheng et al., 2024; Ye et al., 2024; Wang et al., 2024b; Liu et al., 2024; Su et al., 2023) across the 10 TVBench tasks. In addition, we also report a human baseline to verify the quality of the benchmark, more details see appendix A.2.1. We also include the average performance of these models on MVBench and TVBench, with the upward arrow $\uparrow$ indicating the improvement over random chance.

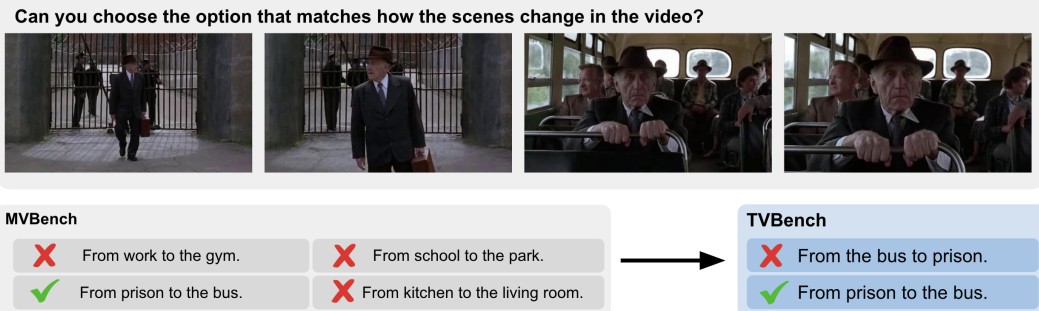

Figure 5: **Strategy 1 of TVBench: Define temporally hard answer candidates.** To address the spatial bias in MVBench, we design answer candidates that are temporally challenging.

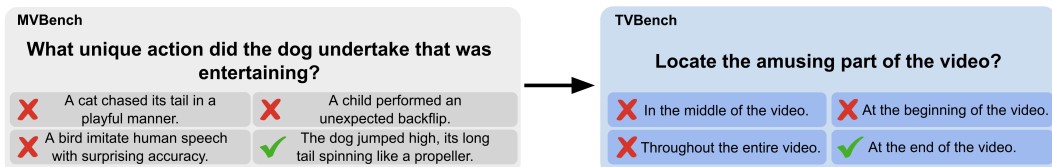

Figure 6: **Strategy 2 of TVBench: Define questions that are not overly informative.** To address the textual bias in MVBench, we design QA templates that do not contain unnecessary information.

**Does time matter?** For TVBench, multi-modal LLMs with a single image perform at random chance, verifying that a random frame is not sufficient for accurate question answering. Specifically, Gemini 1.5 Pro, the top image-language model on TVBench, outperforms random chance by only 3.0%, compared to a 21.2% improvement on MVBench. Shuffling videos has minimal impact on the performance of video-language models on MVBench, but significantly degrades their accuracy on TVBench, where it drops to near-random levels. For example, Tarsier-34B's accuracy is 33.9% higher than the random baseline on MVBench when videos are shuffled, while on TVBench, it is only 4.7% higher under the same conditions. This suggests that temporal understanding is crucial for TVBench, where visual data alone is insufficient to outperform random chance, unlike MVBench. In addition, we analyze the effect of reversing videos instead of shuffling them. This significantly degrades model performance on TVBench, resulting in accuracy below the random baseline for even the best models. For instance, Tarsier-7B and Tarsier-34B perform worse than random by 6.1% and 4.9%, respectively. This is expected, as top models on TVBench rely on temporal information. Reversing frames misleads them with incorrect cues while shuffling disrupts the temporal structure entirely. These results demonstrate that temporal understanding is crucial for TVBench which is in contrast to MVBench where reversed and non-reversed videos obtain the same performance.

**Does vision matter?** For TVBench, state-of-the-art LLMs with text-only perform at random levels, highlighting the effectiveness of our Strategy 2 for Problem 2. Notably, Llama 3 achieves the best performance, just 1.4% above random chance on TVBench, whereas it performs 10.8% better on MVBench. This indicates that LLMs cannot determine the answer solely by analyzing the question and answer candidates or by relying on prior world knowledge. Thus, visual information becomes key for solving TVBench.

# 6 DISCUSSION

**A sobering view on current models.** With our new TVBench, we can accurately assess the temporal understanding of existing video-language models. Surprisingly, we find that recent state-of-the-art and highly popular models, such as VideoChat2, ST-LLM, PLLava, VideGPT+, GPT-4o, VideoLLaMA2 7B, mPLUG-Owl3 perform close to random chance on our temporal benchmark, with the largest gain being only +8.6%. The Gemini 1.5 Pro, Qwen2-VL, LLaVA-Video, IXC-2.5, and Tarsier models outperform the random baseline, achieving up to +20.5%. The primary distinction between Tarsier and previous open-sourced models lies in its substantial increase in pretraining data, leveraging 3.8 million vision-text samples, of which only 1 million are images. From these results, we observe that TVBench amplifies the performance gaps between models with the strongest temporal understanding and those with weaker capabilities, as a good benchmark should.

**Conclusion.** In this work, we highlight major limitations in existing language-video benchmarks, particularly in the widely used MVBench. Key issues include inadequate temporal evaluation and

Table 4: **Results on TVBench.** On TVBench, text-only and image models perform near-random. Surprisingly, several recent state-of-the-art video-language models, such as ST-LLM, also perform close to random. With TVBench we can identify temporally strong models like Gemini and Tarsier. In addition, these models drop significantly in accuracy when the videos are shuffled or reversed, indicated by the upward arrow $\uparrow$ showing the difference to random chance.

| Model | Input | MVBench Average | TVBench Average | TVBench | | | | | | | | | |
|---|---|---|---|---|---|---|---|---|---|---|---|---|---|
| | | | | AC | OC | AS | OS | ST | AL | AA | UA | ES | MD |
| Random | – | 27.3 | 33.3 | 25.0 | 25.0 | 50.0 | 33.3 | 50.0 | 25.0 | 50.0 | 25.0 | 25.0 | 25.0 |
| GPT-3.5 Turbo | text-only | $35.0_{\uparrow7.7}$ | $33.1_{\downarrow0.2}$ | 27.2 | 18.2 | 44.9 | 32.0 | 53.5 | 26.9 | 45.9 | 29.2 | 26.5 | 26.7 |
| Llama 3 70B | | $38.1_{\uparrow10.8}$ | $34.7_{\uparrow1.4}$ | 30.2 | 27.0 | 48.7 | 32.9 | 55.1 | 26.9 | 49.1 | 23.3 | 25.5 | 28.0 |
| GPT-4o | | $34.8_{\uparrow7.5}$ | $33.8_{\uparrow0.5}$ | 28.0 | 21.0 | 48.7 | 33.3 | 53.5 | 25.6 | 50.9 | 25.8 | 28.5 | 22.4 |
| Gemini 1.5 Pro | | $38.2_{\uparrow10.9}$ | $33.6_{\uparrow0.3}$ | 25.0 | 17.6 | 53.0 | 33.3 | 51.9 | 25.0 | 54.7 | 23.3 | 28.0 | 24.1 |
| Tarsier-34B | | $35.7_{\uparrow8.4}$ | $34.4_{\uparrow1.1}$ | 28.7 | 25.0 | 50.9 | 33.3 | 49.7 | 26.2 | 54.7 | 22.5 | 23.5 | 29.7 |
| Idefics3 | image | $44.2_{\uparrow16.9}$ | $34.5_{\uparrow1.2}$ | 27.1 | 23.0 | 56.8 | 36.9 | 49.2 | 25.6 | 52.2 | 29.2 | 25.5 | 19.8 |
| GPT-4o | | $47.8_{\uparrow20.5}$ | $35.8_{\uparrow2.5}$ | 30.8 | 19.6 | 62.1 | 33.3 | 52.4 | 25.0 | 52.5 | 27.5 | 33.0 | 21.6 |
| Gemini 1.5 Pro | | $48.5_{\uparrow21.2}$ | $36.3_{\uparrow3.0}$ | 22.8 | 21.0 | 55.9 | 36.4 | 51.4 | 36.9 | 54.7 | 35.8 | 30.0 | 23.7 |
| Tarsier-34B | | $45.1_{\uparrow17.8}$ | $35.0_{\uparrow1.7}$ | 30.0 | 26.4 | 61.0 | 27.1 | 53.0 | 32.5 | 49.4 | 27.5 | 21.5 | 22.0 |
| VideoChat2 | video reverse | $46.7_{\uparrow19.4}$ | $32.1_{\downarrow1.2}$ | 28.2 | 22.3 | 50.6 | 33.3 | 38.4 | 23.1 | 45.6 | 33.3 | 23.0 | 22.8 |
| ST-LLM | | $53.4_{\uparrow26.1}$ | $34.7_{\uparrow1.4}$ | 25.4 | 32.4 | 55.1 | 36.9 | 43.6 | 30.5 | 47.5 | 31.4 | 23.5 | 20.3 |
| PLLaVA-7B | | $45.8_{\uparrow18.5}$ | $34.1_{\uparrow0.8}$ | 32.6 | 29.1 | 53.4 | 33.3 | 48.6 | 24.4 | 48.1 | 28.3 | 22.0 | 21.6 |
| PLLaVA-13B | | $48.4_{\uparrow21.1}$ | $34.6_{\uparrow1.3}$ | 36.9 | 27.7 | 51.9 | 33.3 | 52.4 | 25.0 | 46.6 | 34.2 | 19.5 | 19.0 |
| PLLaVA-34B | | $56.2_{\uparrow28.9}$ | $33.4_{\uparrow0.1}$ | 26.1 | 27.7 | 51.7 | 35.1 | 33.0 | 26.9 | 54.1 | 29.2 | 26.5 | 23.7 |
| Gemini 1.5 Pro | | $53.1_{\uparrow25.8}$ | $27.0_{\downarrow6.3}$ | 29.6 | 16.9 | 45.1 | 32.7 | 24.7 | 22.7 | 26.9 | 39.8 | 23.5 | 7.7 |
| Tarsier-7B | | $62.5_{\uparrow35.2}$ | $28.4_{\downarrow4.9}$ | 24.3 | 19.6 | 41.1 | 36.0 | 38.9 | 24.4 | 29.4 | 30.0 | 24.5 | 15.9 |
| Tarsier-34B | | $67.7_{\uparrow40.4}$ | $27.2_{\downarrow6.1}$ | 30.2 | 25.0 | 45.1 | 32.9 | 31.9 | 21.9 | 19.1 | 38.3 | 23.0 | 4.3 |
| VideoChat2 | video shuffle | $49.8_{\uparrow22.5}$ | $34.7_{\uparrow1.4}$ | 25.6 | 27.0 | 54.0 | 32.9 | 56.2 | 23.1 | 48.1 | 33.3 | 24.5 | 22.4 |
| ST-LLM | | $53.9_{\uparrow26.6}$ | $35.0_{\uparrow1.7}$ | 25.6 | 35.1 | 50.8 | 36.9 | 49.2 | 31.5 | 45.6 | 32.4 | 23.5 | 19.4 |
| PLLaVA-7B | | $46.5_{\uparrow19.2}$ | $34.4_{\uparrow1.1}$ | 34.0 | 29.1 | 51.5 | 33.3 | 49.7 | 24.4 | 49.7 | 28.3 | 22.5 | 21.6 |
| PLLaVA-13B | | $49.5_{\uparrow22.2}$ | $35.1_{\uparrow1.8}$ | 36.8 | 25.0 | 55.5 | 33.3 | 51.9 | 27.5 | 45.9 | 35.0 | 21.0 | 19.0 |
| PLLaVA-34B | | $56.7_{\uparrow29.4}$ | $37.2_{\uparrow3.9}$ | 25.9 | 29.1 | 58.5 | 35.1 | 56.2 | 34.4 | 55.9 | 28.3 | 25.0 | 23.7 |
| Gemini 1.5 Pro | | $56.8_{\uparrow29.5}$ | $36.1_{\uparrow2.8}$ | 26.5 | 23.7 | 55.9 | 35.1 | 51.4 | 31.3 | 51.9 | 31.7 | 29 | 24.6 |
| Tarsier-7B | | $56.9_{\uparrow29.6}$ | $36.0_{\uparrow2.7}$ | 21.1 | 38.5 | 54.5 | 38.2 | 53.5 | 30.0 | 55.6 | 27.5 | 24.0 | 16.8 |
| Tarsier-34B | | $61.2_{\uparrow33.9}$ | $38.0_{\uparrow4.7}$ | 30.2 | 35.8 | 59.8 | 34.7 | 52.4 | 35.6 | 55.6 | 35.0 | 23.0 | 18.1 |
| VideoLLaVA | video | $42.5_{\uparrow15.2}$ | $33.8_{\uparrow0.5}$ | 26.3 | 27.0 | 54.7 | 33.8 | 57.8 | 23.1 | 45.7 | 24.1 | 22.5 | 23.3 |
| VideoChat2 | | $51.0_{\uparrow23.7}$ | $33.0_{\downarrow0.3}$ | 25.9 | 21.6 | 51.7 | 33.3 | 48.1 | 23.1 | 44.7 | 34.2 | 25.0 | 22.8 |
| ST-LLM | | $54.9_{\uparrow27.6}$ | $35.3_{\uparrow2.0}$ | 25.0 | 35.1 | 49.2 | 36.0 | 54.4 | 31.0 | 45.6 | 32.4 | 24.0 | 20.3 |
| GPT-4o | | $49.1_{\uparrow21.8}$ | $39.1_{\uparrow5.8}$ | 26.1 | 21.3 | 59.7 | 33.2 | 52.4 | 25.0 | 78.4 | 33.3 | 31.0 | 30.6 |
| PLLaVA-7B | | $46.6_{\uparrow19.3}$ | $34.2_{\uparrow0.9}$ | 32.1 | 25.7 | 52.1 | 33.3 | 52.4 | 23.8 | 53.1 | 27.5 | 20.5 | 21.6 |
| PLLaVA-13B | | $50.1_{\uparrow22.8}$ | $35.5_{\uparrow2.2}$ | 37.3 | 24.3 | 57.6 | 33.3 | 55.1 | 28.1 | 47.8 | 35.0 | 19.5 | 17.2 |
| PLLaVA-34B | | $58.1_{\uparrow30.8}$ | $41.9_{\uparrow8.6}$ | 27.6 | 32.4 | 64.8 | 35.6 | 77.8 | 44.4 | 58.8 | 31.7 | 27.0 | 19.4 |
| mPLUG-Owl3 | | $54.5_{\uparrow27.2}$ | $41.4_{\uparrow8.1}$ | 27.4 | 32.4 | 68.2 | 37.3 | 76.8 | 43.1 | 56.9 | 27.5 | 18.0 | 26.3 |
| VideoLLaMA2.1 | | $57.3_{\uparrow30.0}$ | $41.4_{\uparrow8.1}$ | 25.4 | 30.4 | 70.1 | 29.8 | 76.6 | 46.9 | 58.1 | 30.8 | 23.5 | 22.4 |
| VideoLLaMA2 7B | | $54.6_{\uparrow27.3}$ | $41.0_{\uparrow7.7}$ | 30.6 | 37.8 | 69.3 | 33.8 | 68.1 | 40.6 | 56.9 | 25.8 | 24.5 | 22.8 |
| VideoLLaMA2 72B | | $62.0_{\uparrow34.7}$ | $47.5_{\uparrow14.2}$ | 26.7 | 46.6 | 72.0 | 40.4 | 81.6 | 53.1 | 76.6 | 29.1 | 19.5 | 29.7 |
| VideoGPT+ | | $58.7_{\uparrow31.4}$ | $41.5_{\uparrow8.2}$ | 30.6 | 52.0 | 60.2 | 36.9 | 69.2 | 40.0 | 53.1 | 28.3 | 16.0 | 28.5 |
| Gemini 1.5 Pro | | $60.5_{\uparrow33.2}$ | $46.5_{\uparrow13.2}$ | 33.5 | 22.3 | 70.4 | 34.5 | 82.6 | 51.6 | 77.8 | 33.3 | 25.0 | 33.6 |
| Qwen2-VL 7B | | $67.0_{\uparrow39.7}$ | $43.6_{\uparrow10.3}$ | 27.1 | 61.5 | 62.5 | 38.7 | 75.7 | 41.3 | 63.1 | 21.7 | 22.5 | 22.4 |
| Qwen2-VL 72B | | $73.6_{\uparrow46.3}$ | $52.5_{\uparrow19.2}$ | 32.6 | 73.0 | 68.4 | 31.1 | 82.2 | 45.0 | 75.3 | 35.8 | 19.5 | 62.1 |
| LLaVA-Video 7B | | $58.6_{\uparrow31.3}$ | $45.2_{\uparrow11.9}$ | 32.6 | 23.6 | 77.7 | 32.9 | 81.6 | 58.8 | 71.9 | 26.7 | 22.5 | 24.1 |
| LLaVA-Video 72B | | $64.1_{\uparrow36.8}$ | $49.6_{\uparrow16.3}$ | 38.6 | 27.0 | 79.5 | 33.3 | 85.9 | 65.6 | 66.6 | 32.5 | 25.5 | 41.8 |
| IXC-2.5-7B | | $69.1_{\uparrow41.8}$ | $50.5_{\uparrow17.2}$ | 32.1 | 50.7 | 75.9 | 37.3 | 78.4 | 46.9 | 60.0 | 32.5 | 21.0 | 70.3 |
| Aria | | $69.7_{\uparrow42.4}$ | $50.5_{\uparrow17.2}$ | 58.4 | 60.1 | 73.9 | 42.2 | 81.6 | 43.1 | 70.3 | 29.2 | 21.0 | 25.0 |
| Tarsier-7B | | $62.6_{\uparrow35.3}$ | $45.8_{\uparrow12.5}$ | 22.9 | 58.8 | 70.1 | 35.6 | 64.9 | 46.9 | 75.6 | 32.5 | 25.5 | 25.0 |
| Tarsier-34B | | $67.6_{\uparrow40.3}$ | $53.8_{\uparrow20.5}$ | 31.7 | 64.2 | 77.1 | 32.1 | 77.8 | 58.1 | 84.4 | 40.0 | 24.5 | 48.3 |
| Human Baseline | | - | $94.8_{\uparrow61.5}$ | 100.0 | 94.9 | 100.0 | 90.6 | 90.0 | 96.0 | 100.0 | 86.0 | 90.0 | 100.0 |

tasks that do not require visual information, making it ineffective for tracking progress in this domain. To address these problems, we introduce TVBench, a benchmark designed to explicitly assess the temporal understanding of video-language models. Our experiments reveal that on TVBench, text-only and visual models lacking temporal reasoning perform at random levels, and only few current models achieve moderately high scores, showing the potential for progress. TVBench thus provides a reliable yardstick for evaluating future advancements in video-language models.

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

## A APPENDIX

### A.1 REPRODUCIBILITY STATEMENT

To ensure the reproducibility of our results, we will make the full dataset publicly accessible in raw form, along with comprehensive documentation detailing its structure and usage. Additionally, we will provide open-sourced evaluation code under a permissive license, enabling researchers to replicate our experiments and adapt the tools for their own studies.

### A.2 DETAILS ON TVBENCH

#### A.2.1 BENCHMARK CREATION

Table 5 provides the template used to generate the QA pairs for each task on TVBench. For the tasks *Action Count*, *Object Count*, *Object Shuffle*, *Action Localization*, *Unexpected Action*, and *Moving Direction*, we use the same set of options for every question. For *Action Sequence* and Scene Transition, we generate candidates by selecting an action and a scene from the video to create negative candidates. The *Action Antonym* candidate set consists of two similar actions that are difficult to distinguish without time understanding, such as 'Wear a jacket' and 'Take off a jacket.' Finally, for *Egocentric Sequence*, we generate random modifications to the correct sequence of actions to create three negative candidates. Table 5 also details the datasets from which videos for each task were sourced. Here we give more details for each task and their template:

**Action Antonym.** We source videos from the NTU (Liu et al., 2019) RGB+D 120 (action classification dataset). Videos are filtered to retain only those belonging to one of 16 selected categories (e.g., take off bag, put on bag, putting something into a bag, take something out of a bag). For each QA, we select the correct option as the label and the alternative option as the opposite action (e.g., take off bag vs put on bag).

**Action Count.** Questions are sourced from the Perception (Patraucean et al., 2024) dataset. QAs are filtered to always include the same candidate set with four options, each being correct for 25% of the questions.

**Action Localization.** Questions are sourced from MVBench(Li et al., 2024b) The candidate set is balanced, with four options, each being correct 25% of the time.

**Action Sequence.** Videos are selected from the STAR (Wu et al., 2024) dataset, containing two actions. The question is always: "What did the person do first?" Two candidates are provided, each representing one of the actions depicted in the video.

**Egocentric Sequence.** We ask for the correct order of tasks performed in egocentric videos taken from CSV (Qian et al., 2022). The question is: "What is the sequence of actions shown in the video?" The correct order is taken from the original annotations. Three negative candidates are generated by swapping two tasks in the correct order.

**Moving Direction.** Videos and annotations are taken from CLEVRER(Yi et al., 2019), including annotations for the positions and directions of each object in the video. The question is: "Which direction does the {yellow sphere} move in the video?" Candidates are always the same four options, each being correct for 25% of the questions: 1. Down and to the left. 2. Down and to the right. 3. Up and to the right. 4. Up and to the left.

**Object Count.** Videos and annotations are taken from CLEVRER(Yi et al., 2019) The question is: "How many moving cylinders are there when the video begins?" The candidate set always consists of four options, each being correct for 25% of the questions. QAs are selected so that ignoring the time constraint (e.g., "when the video begins") would lead to incorrect answers (e.g., the number of objects at the beginning is different from at the end).

**Object Shuffle.** Questions are sourced from Perception(Patraucean et al., 2024) QAs are filtered to always include the same candidate set with four options, each being correct for 25% of the questions.

**Scene Transition.** Questions are sourced from MVBench(Li et al., 2024b) (with videos originally from MoVQA). The original question and correct answer are retained, while incorrect options are discarded. A hard negative is generated by reversing the sequence in the correct answer.

Table 5: Overview of all tasks and datasets used in our TVBench benchmark.

| Task | Source | Question | Candidates |
|---|---|---|---|
| Action Count | Perception | The person makes sets of repeated actions. How many distinct repeated actions did the person do? | 2; 3; 4; 5 |
| Object Shuffle | Perception | The person uses multiple similar objects to play an occlusion game. Where is the hidden object at the end of the game from the person's point of view? | Under the third object from the left. Under the second object from the left. Under the first object from the left. |
| Object Count | CLEVRER | How many {metal} objects are moving when {the video begins}? | 0; 1; 2; 3 |
| Moving Direction | CLEVRER | Which direction does the {gray cube} move in the video? | Down and to the right. Down and to the left. Up and to the right. Up and to the left. |
| Action Sequence | STAR | What did the person do first? | 2 actions that actually happened on the video |
| Scene Transition | MoVQA | What's the right option for how the scenes in the video change? | From {scene1} to {scene2}. From {scene2} to {scene1}. |
| Action Localization | Charades | During which part of the video does the action {person takes a blanket} occur? | Throughout the entire video. At the end of the video. In the middle of the video. At the beginning of the video. |
| Action Antonym | NTU RGB+D | What is the action being performed in the video? | Wear jacket; Take off jacket |
| Unexpected Action | FunQA | Locate the {creative\|amusing\|mesmerizing} part of the video. | Throughout the entire video. At the end of the video. At the beginning of the video. In the middle of the video. |
| Egocentric Sequence | CSV | What is the sequence of actions shown in the video? | GT + Correct sequence changing the order of actions |

Table 6: **TVBench Video Statistics for each task.**

| Task | #Videos | Average Frame Length |
|---|---|---|
| Action Count | 536 | 628 |
| Object Count | 148 | 127 |
| Action Sequence | 528 | 403 |
| Object Shuffle | 225 | 710 |
| Scene Transition | 185 | 599 |
| Action Localization | 160 | 423 |
| Action Antonym | 320 | 171 |
| Unexpected Action | 120 | 735 |
| Egocentric Sequence | 200 | 562 |
| Moving Direction | 232 | 127 |

**Unexpected Action.** Annotations are parsed from the FunQA (Xie et al., 2023) dataset, which provides the start and end times for each action. The question is: "Locate the creative — mesmerizing — surprising part of the video." The candidate set always consists of the following four options: 1. "Throughout the entire video." 2. "At the beginning of the video." 3. "At the end of the video." 4. "In the middle of the video." The correct option is set based on the original annotations.

### A.2.2  HUMAN BASELINE STUDY

With 14 annotators we label 400 videos of the TVBench out of the 2,654 videos in the benchmark. We achieved an average human accuracy of 94.8%.

**Estimating Stastical Error:** By having each annotator annotate 40 different videos, resulting in 400 unique annotations across the dataset, we aim to estimate the human performance baseline with acceptable statistical precision. The total number of videos in the dataset is $N = 2654$, and the number of annotated videos is $n = 400$. Each video is annotated by a single annotator. In the absence of prior knowledge about the true average human accuracy $p$, we adopt a conservative

Table 7: **Analysis of spatial and textual bias on NextQA (Xiao et al., 2021).** Analysis of the NextQA(Xiao et al., 2021) benchmark for spatial and textual bias using the Tarsier model. We find a strong spatial bias, with image performance nearly on par with video, while also having a strong textual bias, text-only version improves significantly on the random baseline.

|  | Input | NextQA |
|---|---|---|
| Random | – | 20.0 |
| Tarsier-34B | text-only | 47.6 |
|  | image | 71.3 |
|  | video shuffle | 78.5 |
|  | video reverse | 77.6 |
|  | video | 79.0 |

approach by assuming $p = 0.5$, which maximizes the variance $p(1 - p)$. This assumption ensures that our estimation of the standard error is not underestimated, providing a robust margin of error. The estimated average accuracy $\hat{p}$ is given by:

$$\hat{p} = \frac{1}{n} \sum_{i=1}^{n} y_i,$$

where $y_i$ is an indicator variable that equals 1 if the annotator answered video $i$ correctly, and 0 otherwise. The variance of $\hat{p}$ with finite population correction is calculated as:

$$\text{Var}(\hat{p}) = \frac{p(1 - p)}{n} \times \left( \frac{N - n}{N - 1} \right).$$

Substituting $p = 0.5$, $n = 400$, and $N = 2654$, we have:

$$p(1 - p) = 0.5 \times 0.5 = 0.25,$$

$$\frac{N - n}{N - 1} = \frac{2654 - 400}{2654 - 1} = \frac{2200}{2599} \approx 0.8496.$$

Therefore, the variance becomes:

$$\text{Var}(\hat{p}) = \frac{0.25}{400} \times 0.8496 = \frac{0.25 \times 0.8465}{400} = \frac{0.211625}{400} = 0.000531.$$

The standard error (SE) is the square root of the variance:

$$\text{SE} = \sqrt{\text{Var}(\hat{p})} = \sqrt{0.000531} \approx 0.0230.$$

At a 95% confidence level (with $z = 1.96$), the margin of error (ME) is:

$$\text{ME} = z \times \text{SE} = 1.96 \times 0.0230 \approx 0.0451.$$

Thus, the statistical error in estimating human performance is characterized by a standard error of approximately 0.0230, leading to a margin of error of $\pm 4.51\%$. This conservative estimate provides an upper bound on the margin of error, ensuring our results are statistically robust even without prior knowledge of $p$.

### A.3 EXTENDED ANALYSIS OF VIDEO MCQA BENCHMARKS

We extend our analysis beyond MVBench to evaluate the NextQA benchmark (Xiao et al., 2021), using the best-performing model, Tarsier-34B. Specifically, we assess performance across five settings: text-only, image-only (random video frame), video, shuffled video, and reversed video. The results, summarized in Table 7, reveal significant spatial and textual biases in the NextQA benchmark. First, we observe a strong spatial bias: the image-only setting achieves a performance nearly on par with the video setting (71.3% vs. 79.0%). Furthermore, altering the temporal structure of videos through shuffling or reversing has minimal impact, with performance dropping by only 0.5% and 1.4%, respectively. Second, the benchmark exhibits a notable textual bias, as many questions

can be answered using text alone. In the text-only setting, Tarsier-34B achieves 47.6%, substantially outperforming the random chance baseline by 27.6%. This highlights significant room for improvement in the benchmark's ability to evaluate genuine temporal reasoning.

### A.4 TVBENCH QUALITATIVE EXAMPLES

This section provides qualitative examples for the different TVBench tasks. As shown in Fig.7-13, temporal information from the input video is indispensable for correctly answering the given questions.

### A.5 SPATIAL AND TEXTUAL BIAS IN MVBENCH

Fig. 14-25 contain several evaluation examples from MVBench that suffer from the issues analyzed in Sec. 3. This qualitative analysis complements the quantitative results discussed in Sec 6 showing that these issues affect a large portion of MVBench.

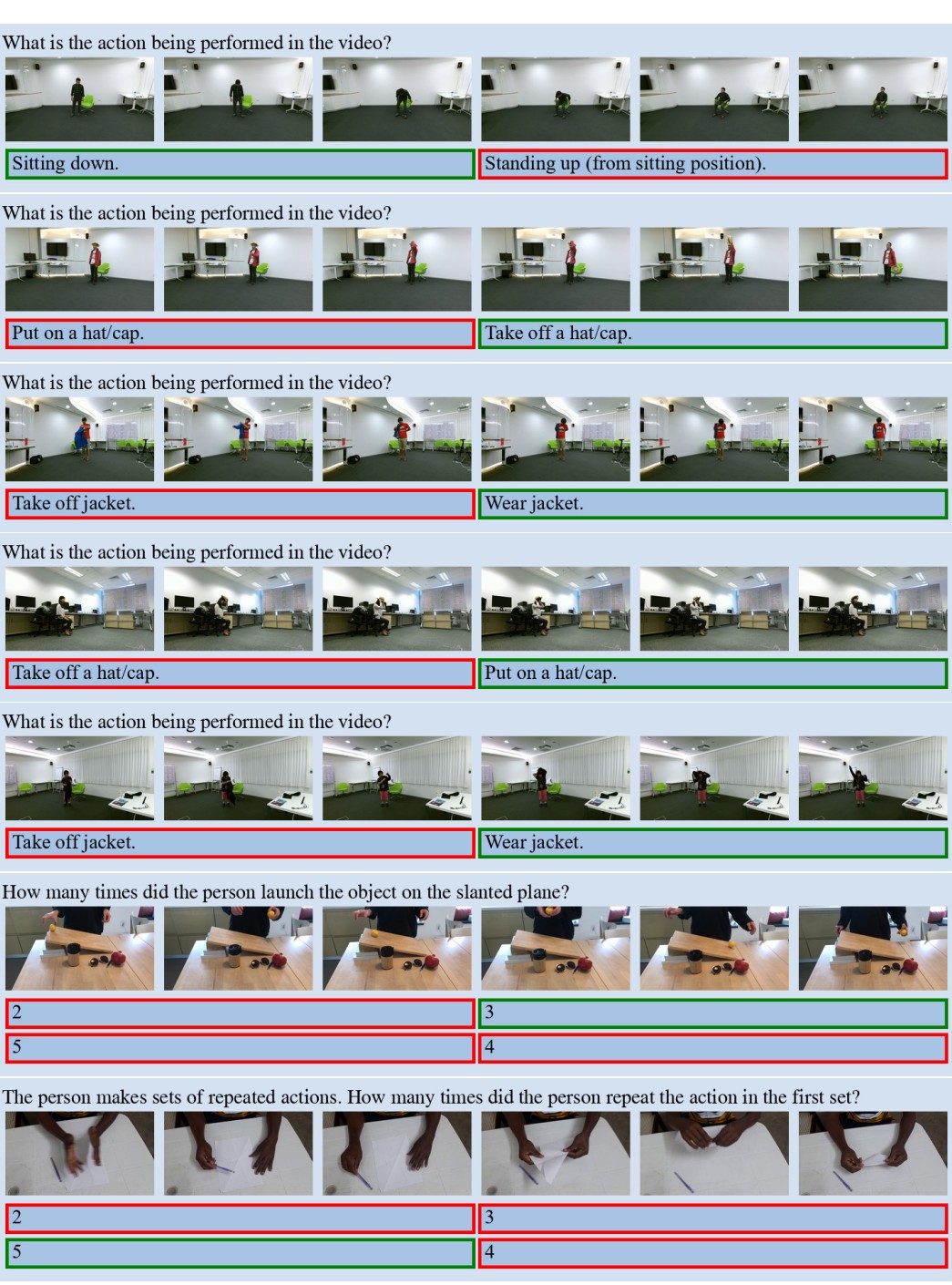

Figure 7: **TVBench: Samples from our benchmark (1).** Row 1-5: Action Antonym; Row 6-7: Action Count.

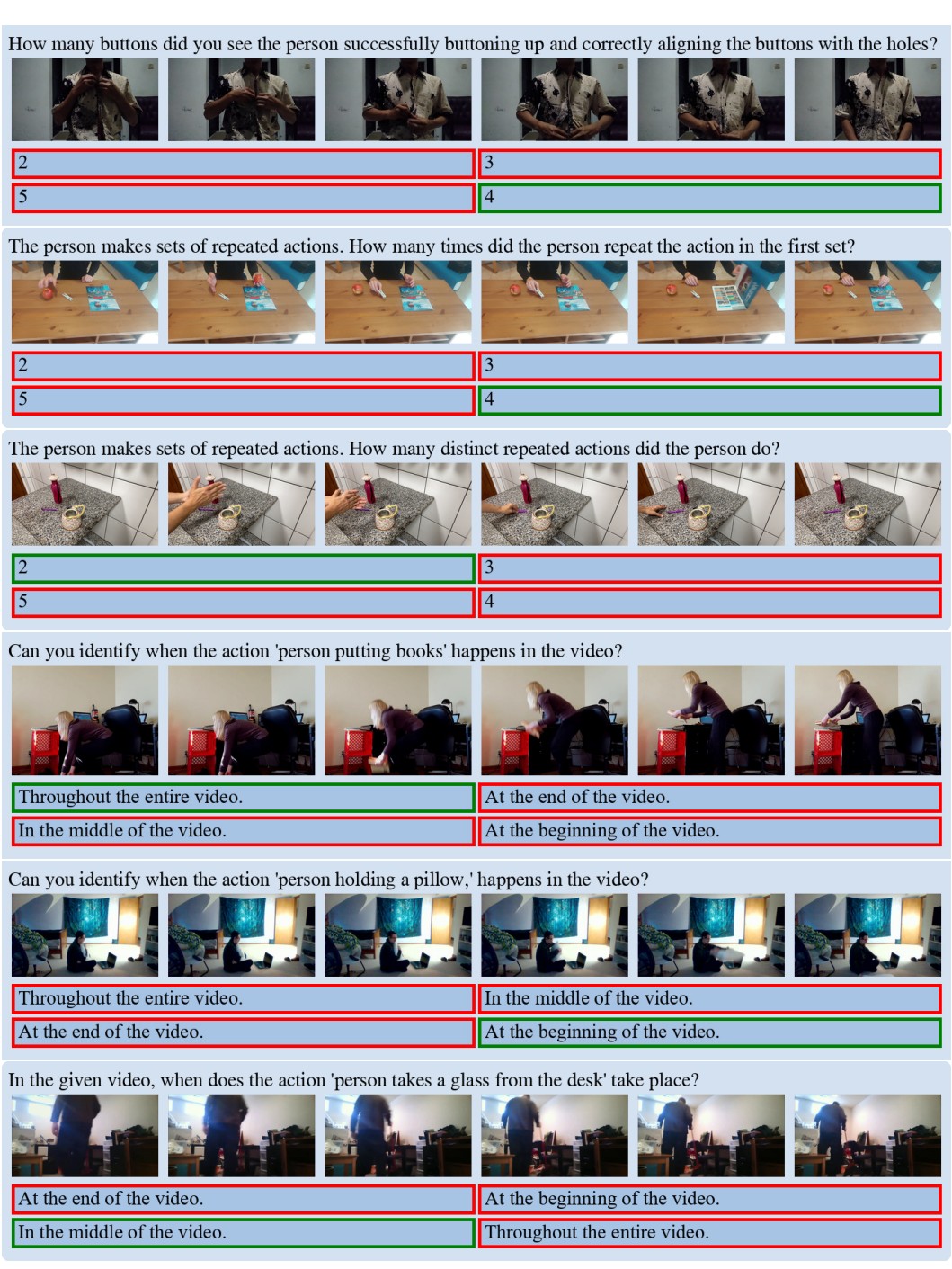

Figure 8: **TVBench: Samples from our benchmark (2).** Row 1-3: Action Count; Row 4-6: Action Localization.

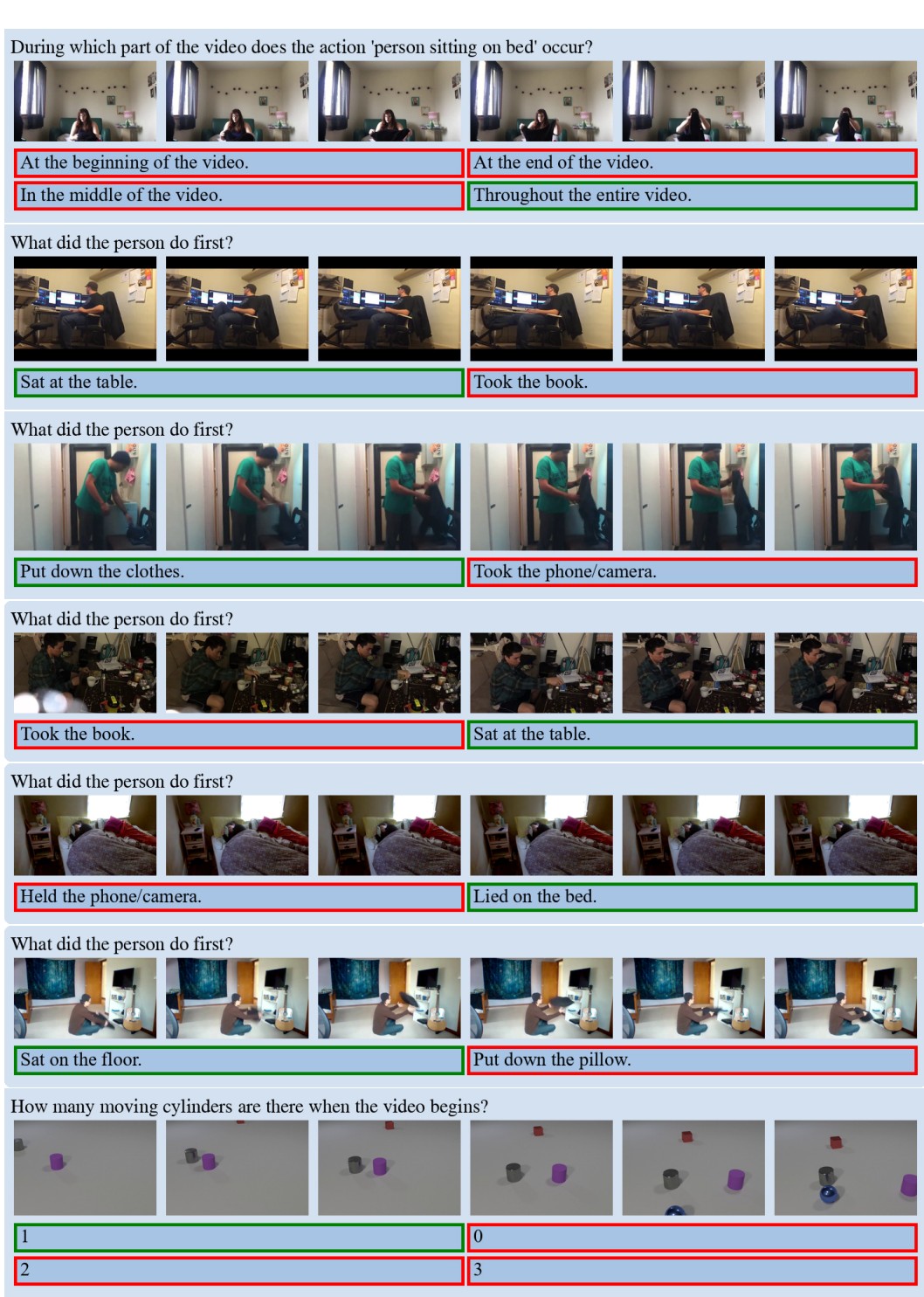

Figure 9: **TVBench: Samples from our benchmark (3).** Row 1: Action Localization; Row 2-6: Action Sequence; Row 7: Object Count.

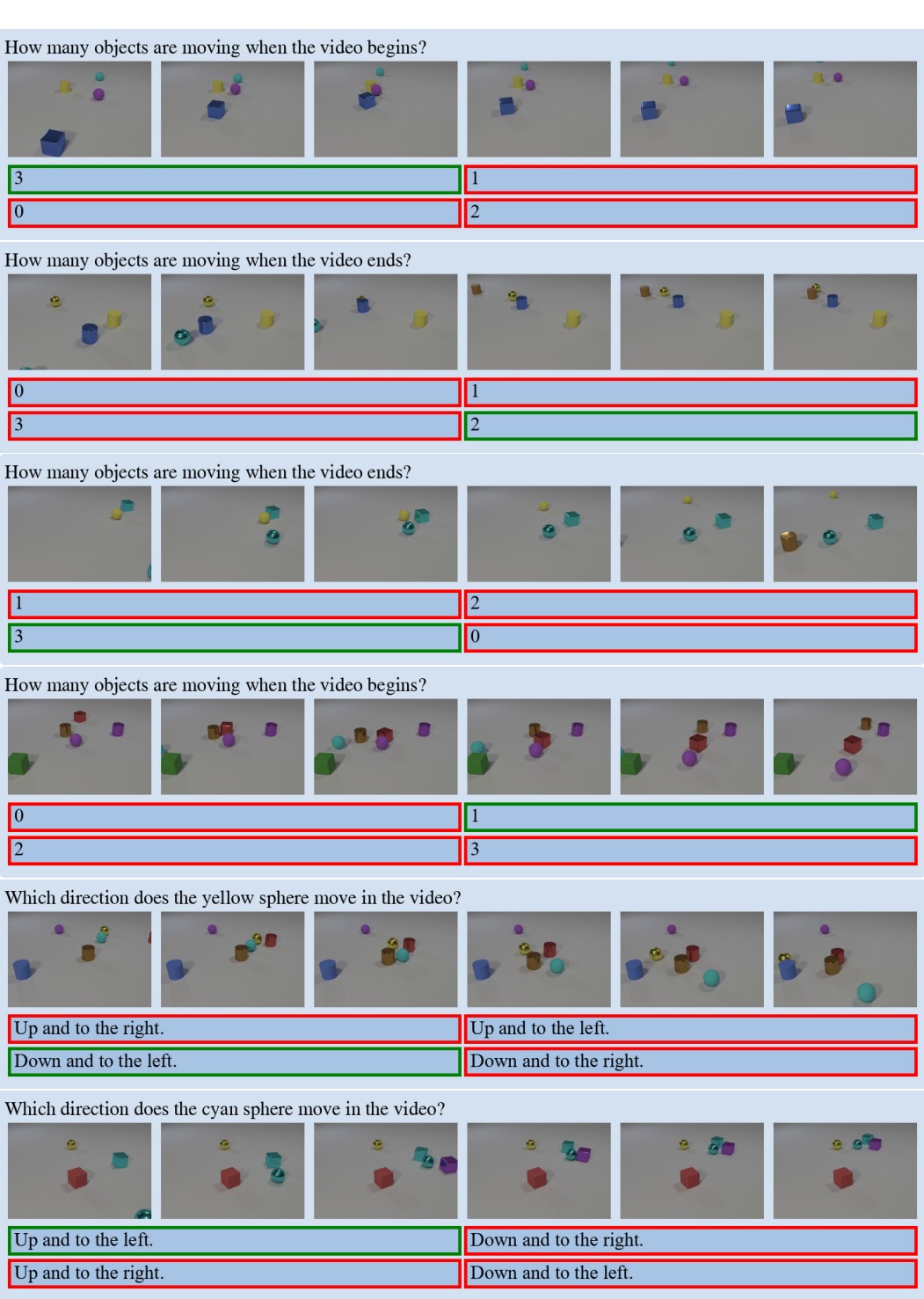

Figure 10: **TVBench: Samples from our benchmark (4).** Row 1-4: Object Count; Row 5-6: Moving Direction.

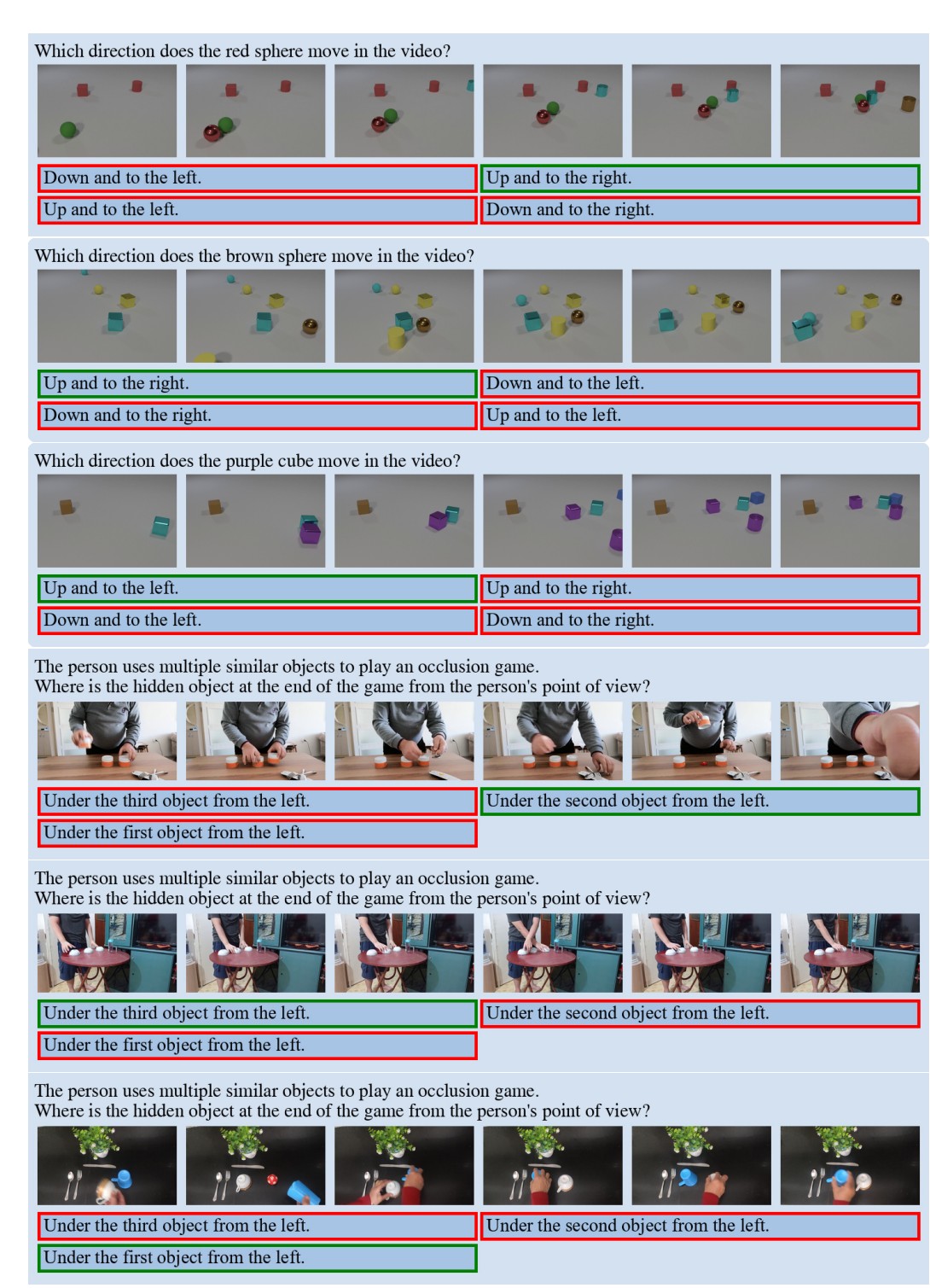

Figure 11: **TVBench: Samples from our benchmark (5).** Row 1-3: Moving Direction; Row 4-6: Object Shuffle.

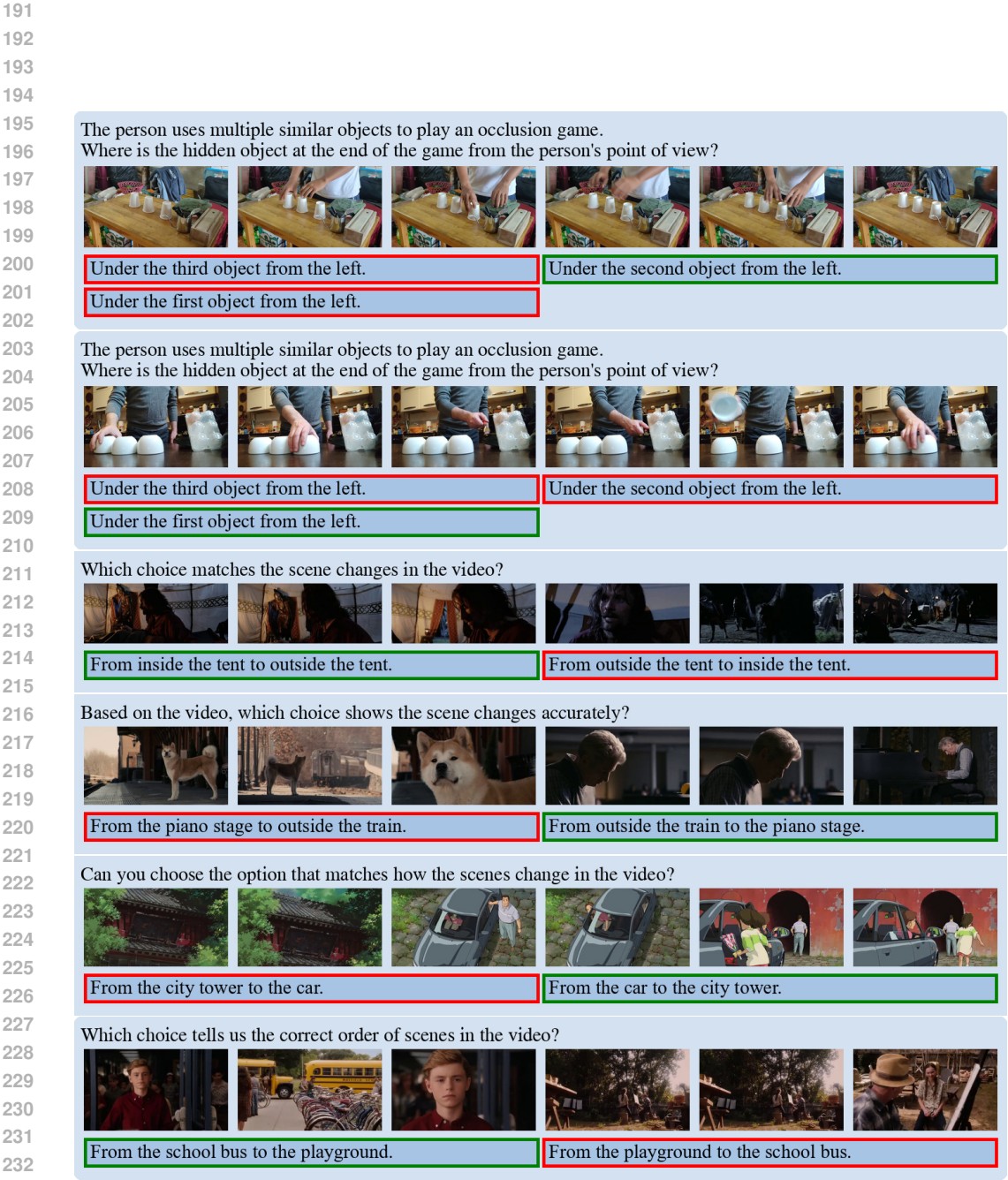

Figure 12: **TVBench: Samples from our benchmark (6).** Row 1-2: Object Shuffle; Row 3-6: Scene Transition.

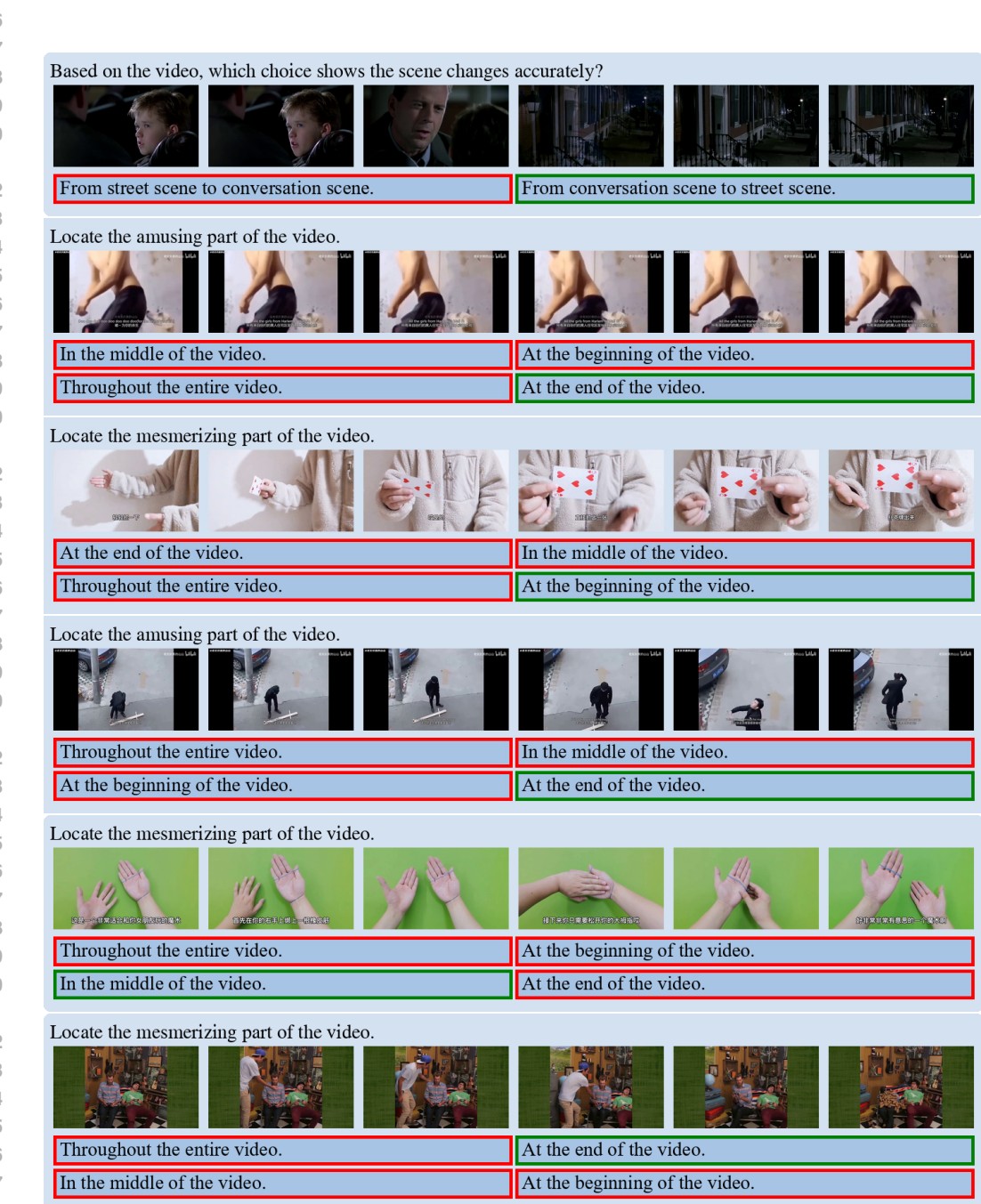

Figure 13: **TVBench: Samples from our benchmark (7).** Row 1: Scene Transition; Row 2-6: Unexpected Action.

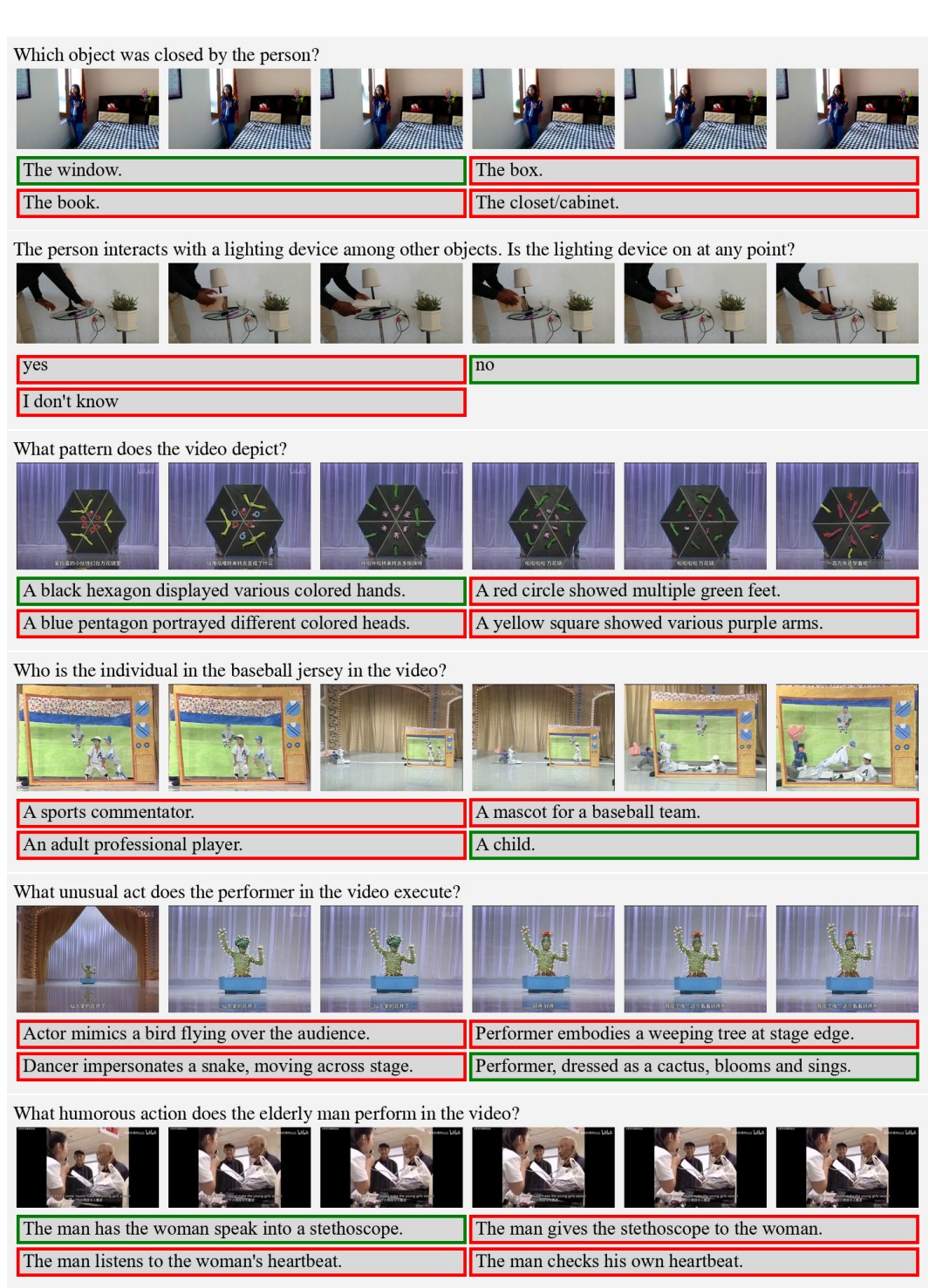

Figure 14: **Spatial bias in MVBench (1).** Multiple-choice questions from various MVBench tasks can be solved using only a single frame from the video.

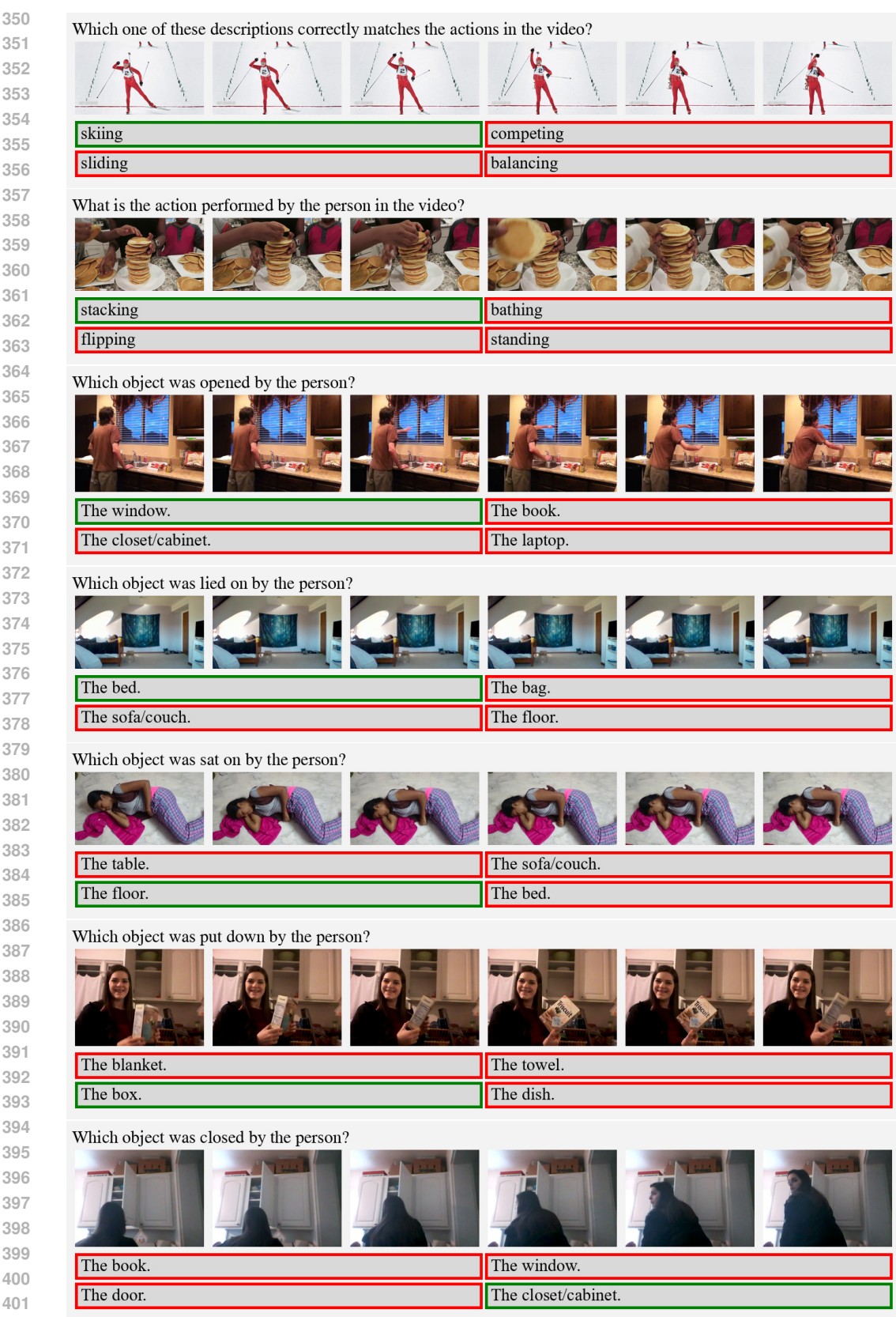

Figure 15: **Spatial bias in MVBench (2).** Multiple-choice questions from various MVBench tasks can be solved using only a single frame from the video.

Figure 16: **Spatial bias in MVBench (3).** A single frame containing all characters inquired is sufficient to answer MVBench.

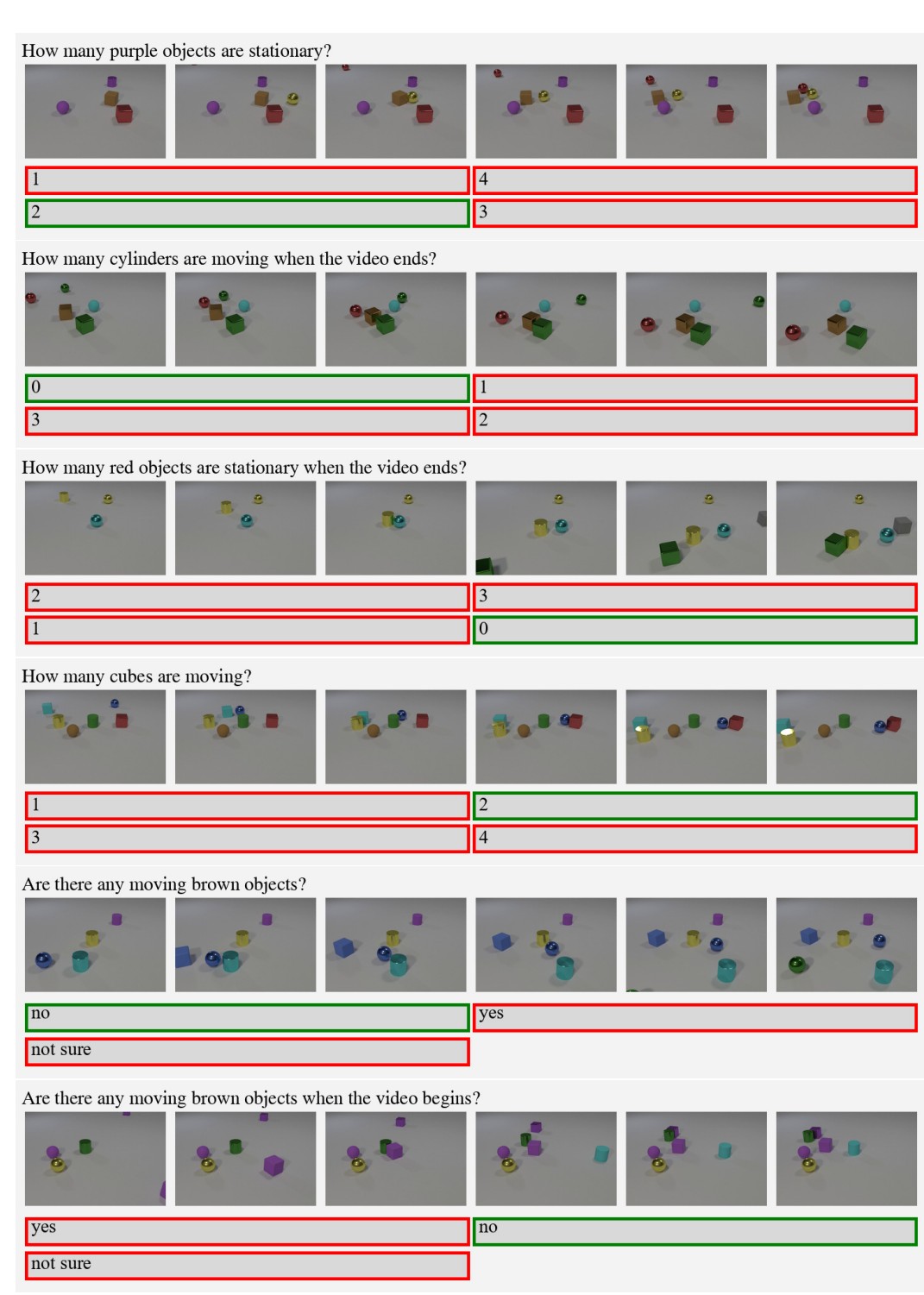

Figure 17: **Spatial bias in MVBench (4).** Temporal understanding is not needed as a single frame suffices to solve these questions on MVBench.

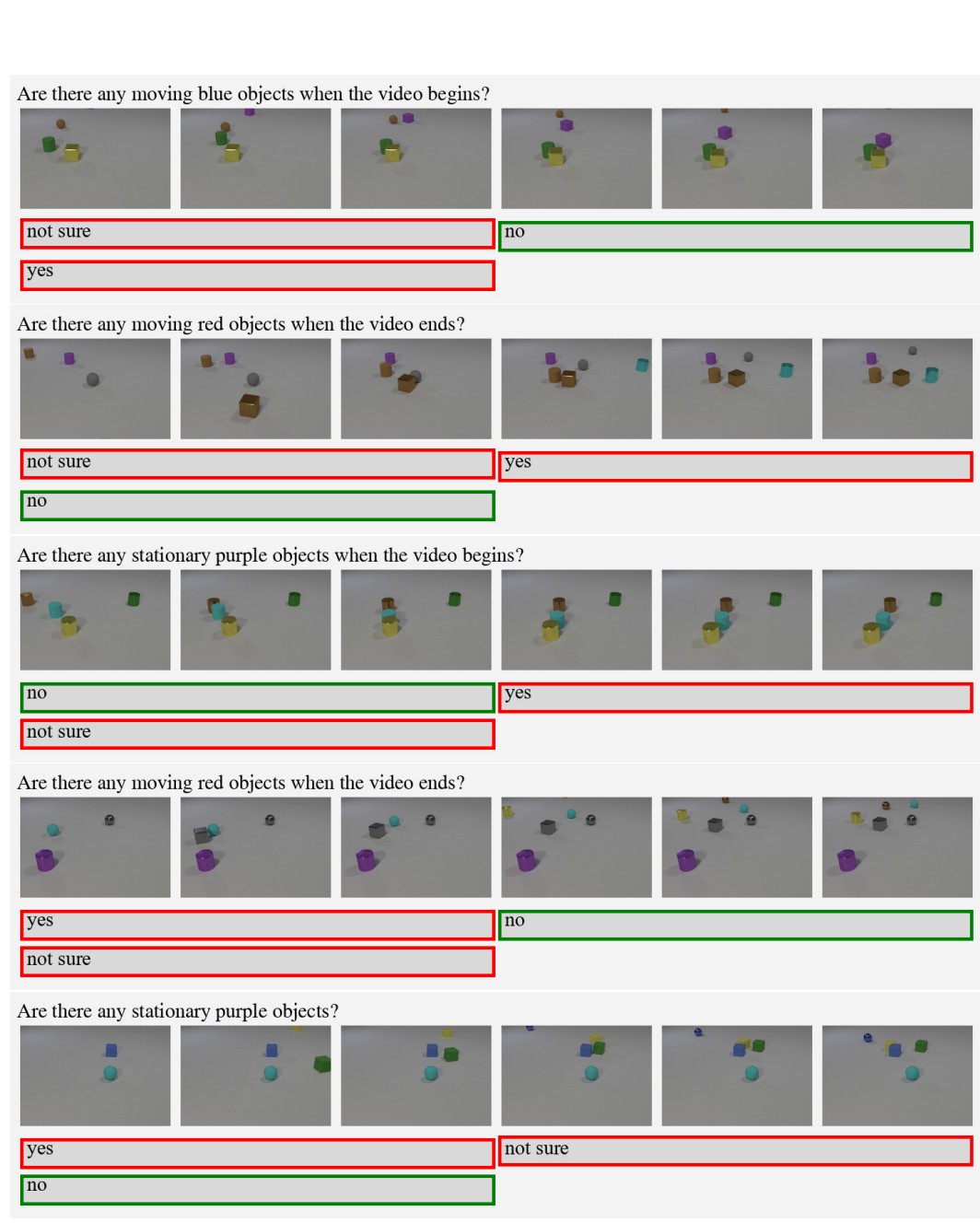

Figure 18: **Spatial bias in MVBench (5).** Temporal understanding is not needed as a single frame suffices to solve these questions on MVBench.

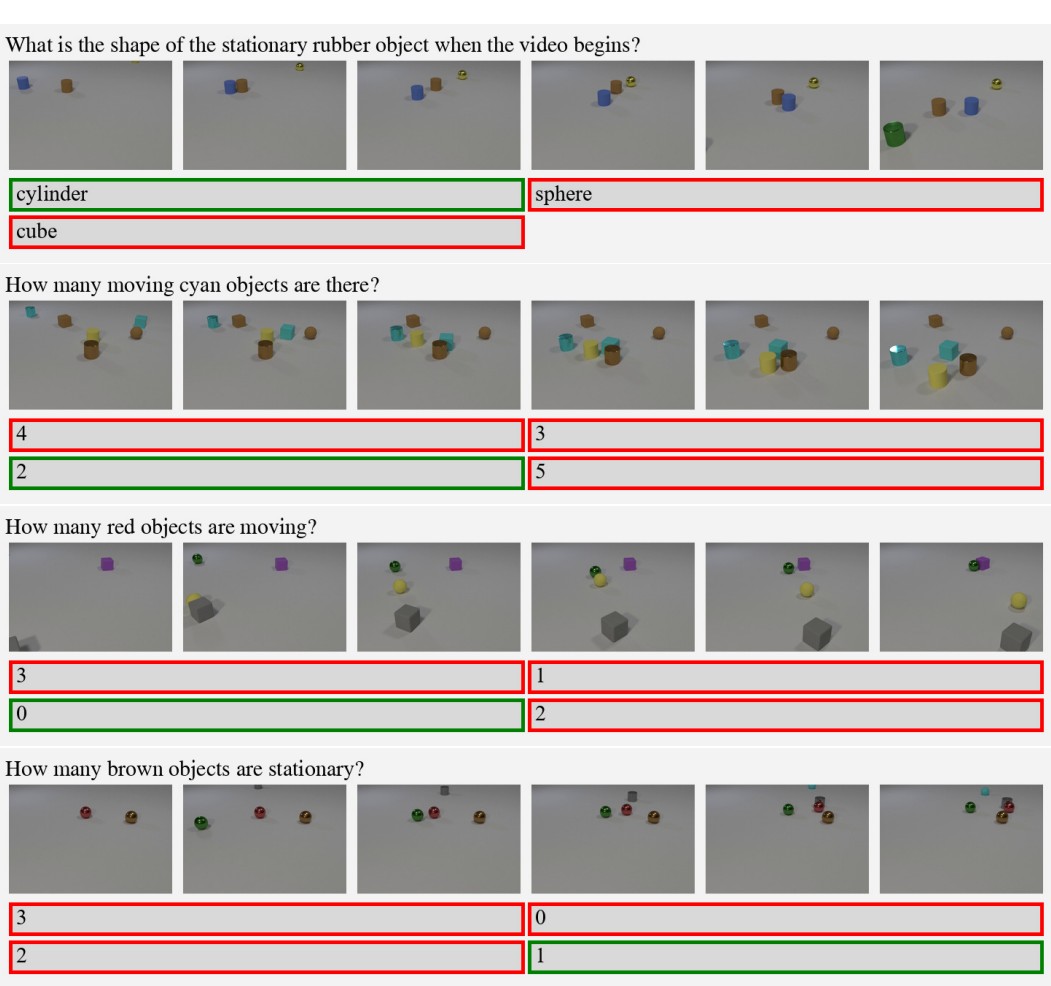

Figure 19: **Spatial bias in MVBench (6).** Temporal understanding is not needed as a single frame suffices to solve these questions on MVBench.

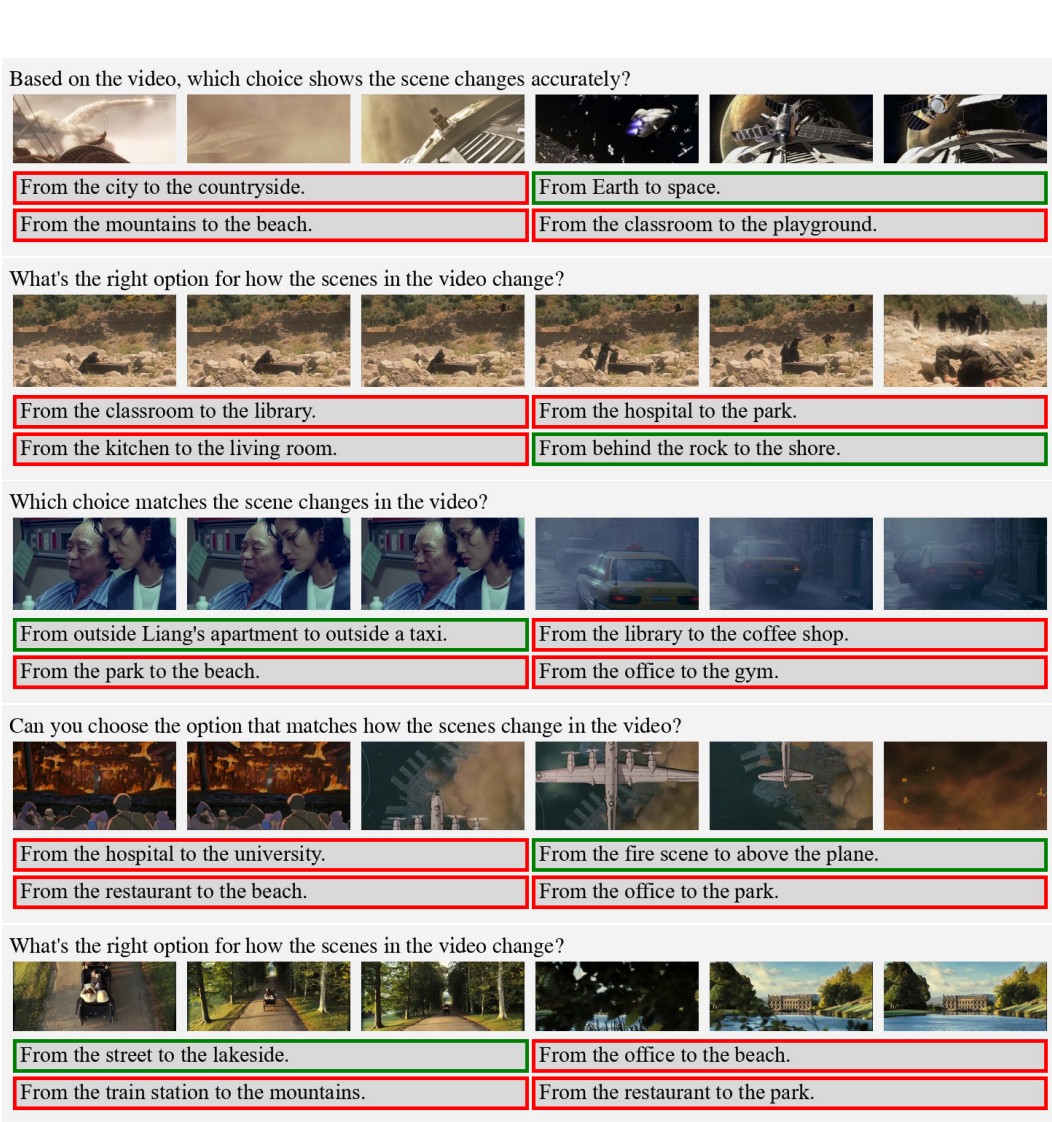

Figure 20: **Spatial bias in MVBench (6).** A single frame is sufficient to discard incorrect candidates, as the scenes described in these candidates never occurred in the video.

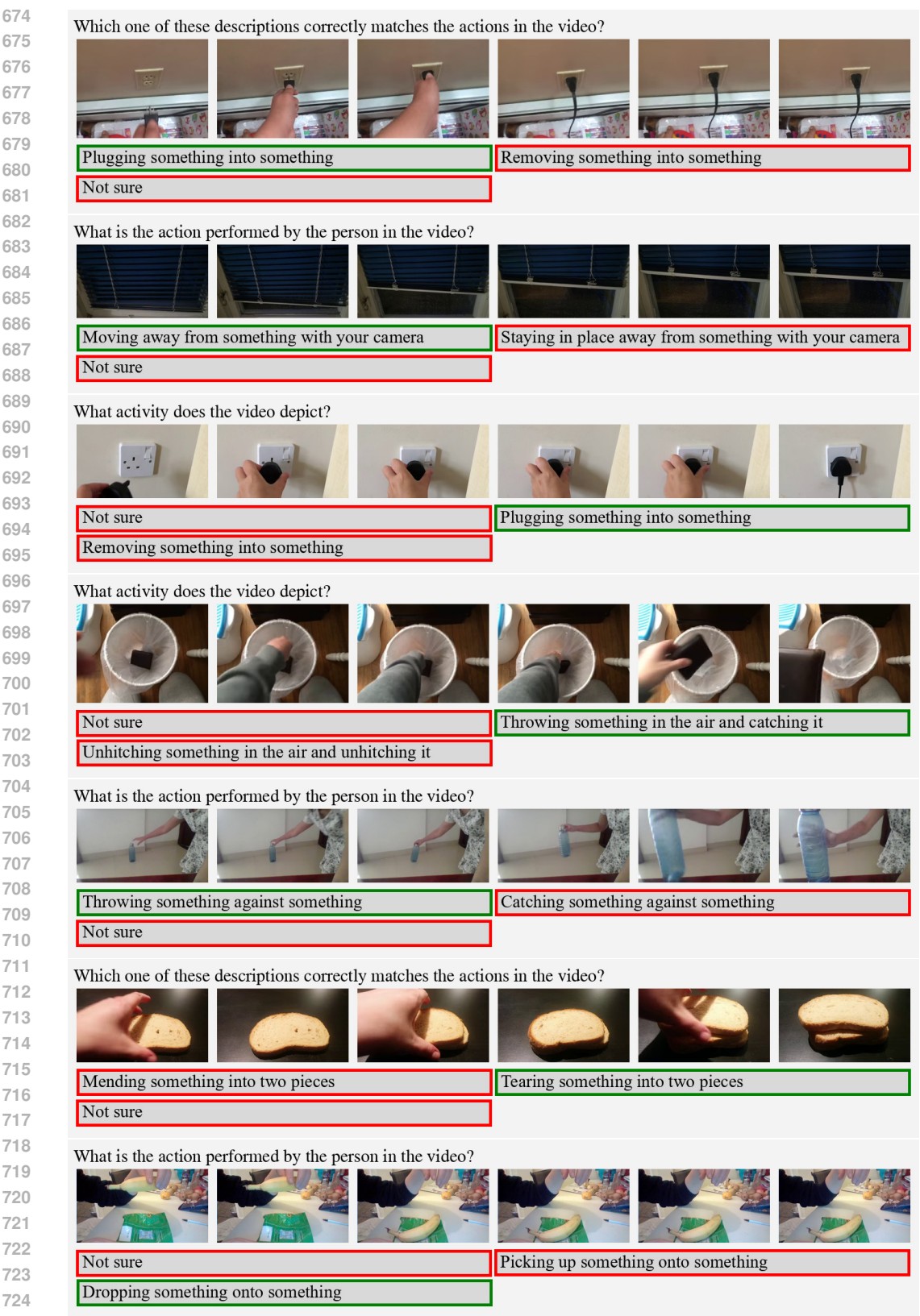

Figure 21: **Textual bias in MVBench (1).** Answer candidates are generated with an LLM. The "Not sure" candidate is never correct, while the other incorrect candidate makes no textual sense e.g. "catching something against something".

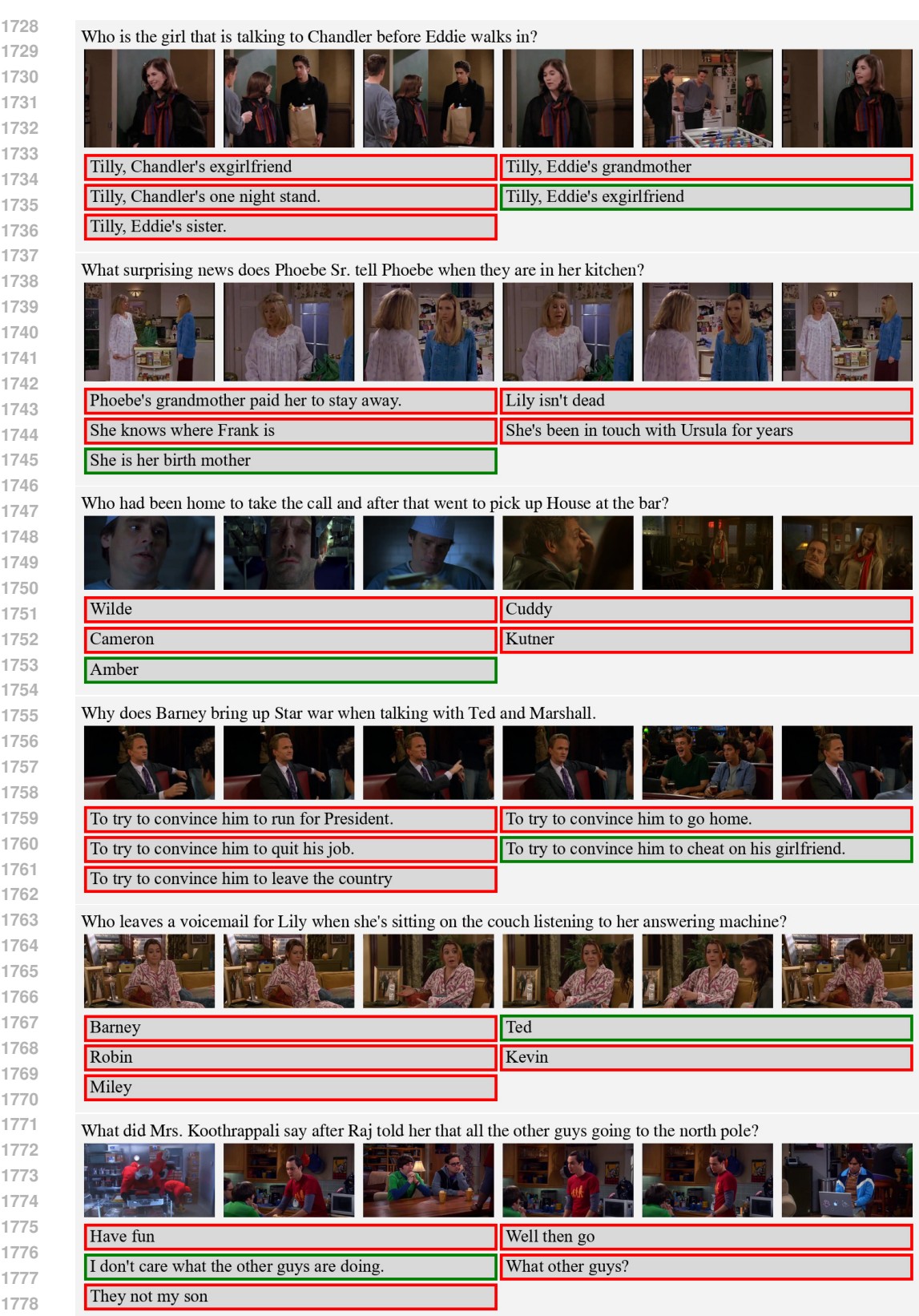

Figure 22: **Textual bias in MVBench (2): Overreliance on world knowledge.** Answers can be inferred using world knowledge of TV shows. Questions cannot be answered from the video only as answers contain e.g. character names of the TV show.

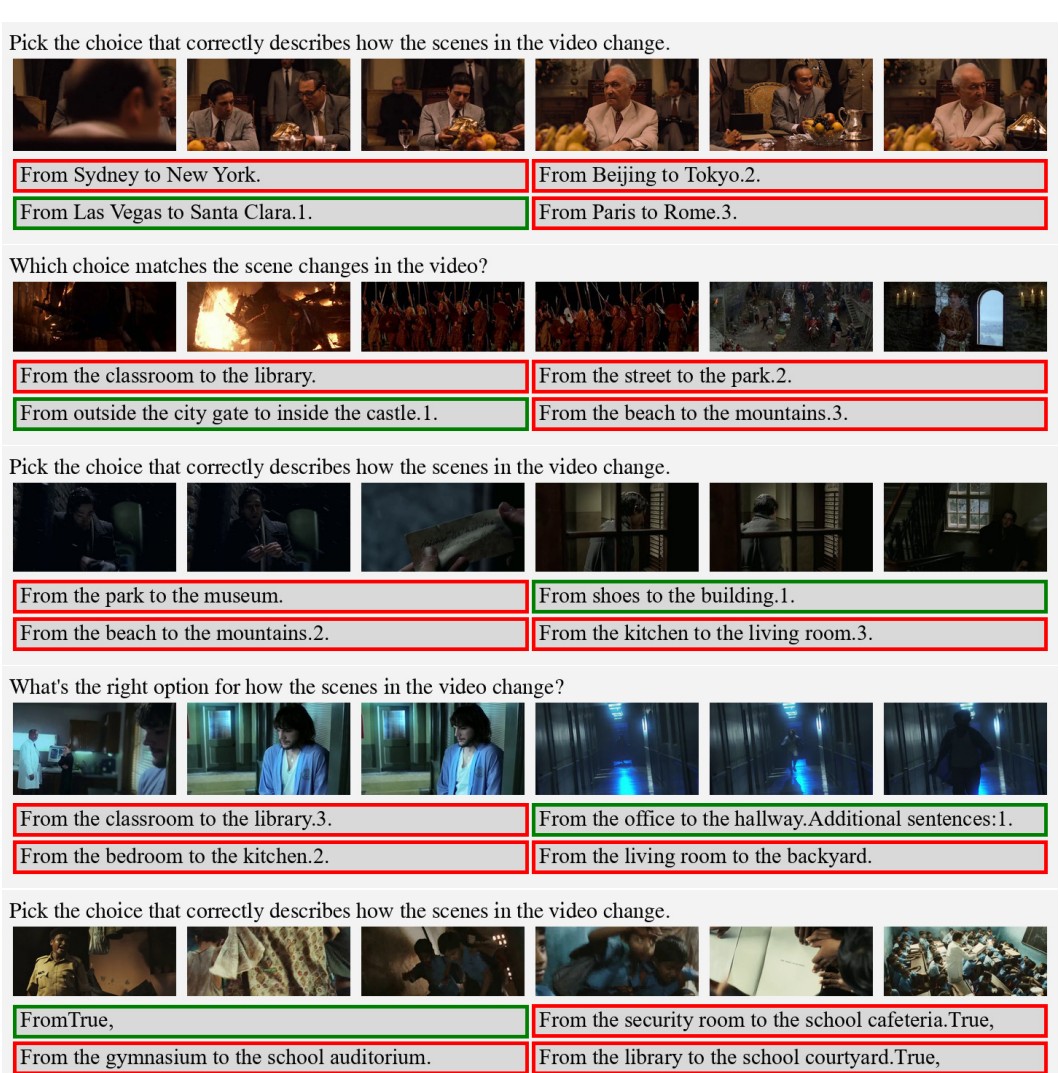

Figure 23: **Textual bias in MVBench (3).** Some candidates in MVBench contain artifacts due to the automatic LLM-based generation process. In these cases, the correct answers end with "1" or contain the keyword "true".

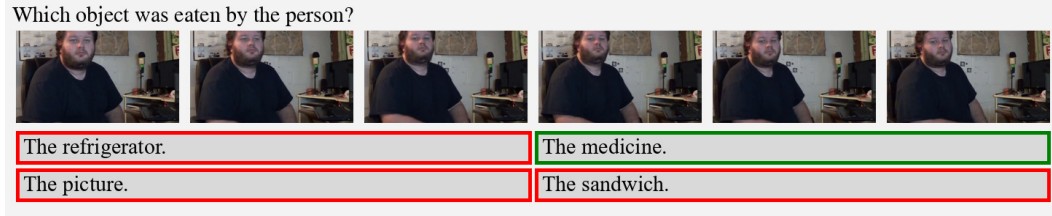

Figure 24: **Textual bias in MVBench (4).** Since two of the candidate objects cannot be eaten, the choice is reduced to two remaining candidates.

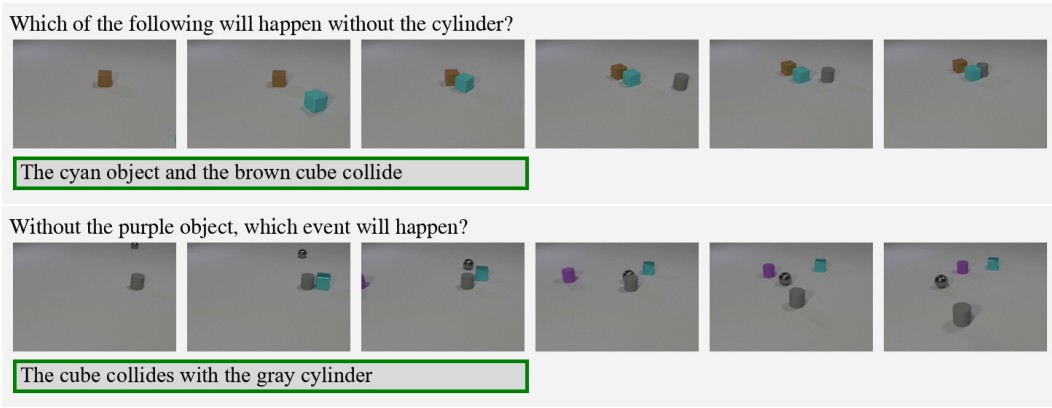

Figure 25: **Textual bias in MVBench (5).** Just one candidate answer is provided for some QA pairs on MVBench.

