# OpenReview forum: "TVBench: Redesigning Video-Language Evaluation"
_ICLR.cc/2025/Conference — Submitted to ICLR 2025_

### Official Review · Reviewer_QEG7 · 2024-11-02

**Soundness:** 3
**Presentation:** 4
**Contribution:** 3
**Rating:** 6
**Confidence:** 4

**Summary:**

This paper analyzed issues in the current most popular video-language benchmark (MVBench), and propose a new benchmark that alleviates the issues. Specifically, the authors provide solid evidence showing MVBench is less temporal, less visual, and simple solution of making an open-ended evaluation protocol can't address the problem. The authors then manually design question types and create questions by combining and filtering questions from existing video-language benchmarks. The authors show the resulting dataset is does not suffer from the issues.

**Strengths:**

- The paper works on an important problem of video-language model evaluation, and identify issues in a widely used benchmark with solid evidences. I believe the impact of the paper will be high.

- I like the analysis and experiments the authors provided for MVBench, the numbers in Table 1 and Table 2 are convincing, and the examples in Figure 2-4 are illustrative. The presentation is well structured and convincing.

- Evaluation on the new benchmark in Table 4 shows text and image only baselines are close to random, and shuffle or reverse the frames drops the performance for all models. This supports that the proposed benchmark emphasizes on temporal information. Table 4 contains results of a wide range of models.

**Weaknesses:**

- While this paper convincingly shows the proposed benchmark is better than MVBench, the authors did not provide discussion/ evidence whether if it is the final video language evaluation benchmark. For example, the benchmark might be too short / in too limited domains / person centric (I am not raising concerns on these particular issues). Some statistics about the datasets are needed.

- The 10 tasks picked in Table 5 looks a bit arbitrary / artificial to me. It will be helpful if the authors provide more rationale why these tasks are picked.

- It is unclear to my how the question-answers are created. Are then all from existing datasets, or the authors hired raters to filter / verify them?

**Questions:**

Overall this paper works on an important problem and provided a valid solution (a new benchmark). The analysis are backed up by solid experiments, and I believe this paper will have positive impact on the community. My concerns are mostly on discussions and clarity, and I expect the authors to address them in the rebuttal. My current rating is a weak accept, and I am happy to raise my rating if my concerns are addressed.

---

> ### Author Response · Authors · 2024-11-21
> **Rebuttal (1/2)**
>
> Thank you for the time and effort spent in reviewing our paper. In the following, we reply to all the concerns raised in the review.
>
> **W1 – Discussion on TVBench and future evaluation benchmarks:** Thank you for highlighting this. We believe that new benchmarks must be created every few years to keep pace with the rapid advancements in AI. However, our benchmark, TVBench, is far from being solved. Even the best-performing method, Tarsier-34B, achieves only ~20% above the random baseline, while many recent models, such as mPLUG-Owl3, GPT-4o, VideoGPT+, and PPLaVA, perform close to random chance, with less than a 9% improvement.
> In TVBench, we source videos for each task from various existing datasets (see Section 5.2), including Perception Test, CLEVRER, STAR, MoVQA, Charades-STA, NTU RGB+D, FunQA, and CSV.
> TVBench thus includes a diverse range of scenarios, featuring both first- and third-person perspectives, indoor and outdoor environments, and real and synthetic data. It comprises 2,654 QA pairs across 10 different tasks, ensuring robust coverage of various temporal challenges. This diversity is essential for creating a benchmark that truly assesses temporal reasoning capabilities. For video QA examples of TVBench, please refer to Appendix A.4.
>
> **W2 – Details on Task Selection:** Our benchmark, TVBench, draws inspiration from existing datasets [1, 2] to cover different skill areas—Memory, Abstraction, Physics, and Semantics—through 10 selected temporally challenging tasks: repetition counting (Action Count); properties of moving objects (Object Shuffle, Object Count, Moving Direction); temporal localization (Action Localization, Unexpected Action); sequential ordering (Action Sequence, Scene Transition, Egocentric Sequence); and distinguishing between similar actions (Action Antonyms). We finalized these 10 tasks by verifying that they are free from spatial and textual biases, unlike previous benchmarks.
>
> --- continued in next comment ---

---

> > ### Author Response · Authors · 2024-11-21
> > **Rebuttal (2/2)**
> >
> > **W3 – Clarification on Benchmark Creation:** Thank you for bringing this up. In TVBench, we carefully design QA templates for each task to ensure they cannot be solved using only text or a single random frame. Rather than hiring annotators, we derive these templates directly from the original dataset annotations. However, we intentionally avoid using the existing QA pairs from these datasets, as they are often unbalanced and prone to textual or spatial biases. To validate the quality of our benchmark, we conducted a human baseline study, achieving a performance of 95%. For more details, please refer to Appendix A.2.2.
> > We have revised Section 5.1 to provide a clearer explanation of the two strategies used in creating TVBench, see below. Furthermore, Appendix A.2.1 offers detailed information about each task, including the templates, questions and answers, and video statistics.
> >
> > ### Strategy 1: Define Temporally Hard Answer Candidates.
> >
> > To address Problem 1, the temporal constraints in the question must be essential for determining the correct answer. This involves designing time-sensitive questions and selecting temporally challenging answer candidates.
> > - We select 10 temporally challenging tasks that require: Repetition counting (Action Count), Properties of moving objects (Object Shuffle, Object Count, Moving Direction), Temporal localization (Action Localization, Unexpected Action), Sequential ordering (Action Sequence, Scene Transition, Egocentric Sequence), distinguishing between similar actions (Action Antonyms).
> > - We define hard-answer candidates based on the original annotations to ensure realism and relevance, rather than relying on LLM-generated candidates that are often random and easily disregarded, as seen in MVBench. For example, in the Scene Transition task (see Figure 6), we design a QA template that provides candidates based on the two scenes occurring in the videos for this task, rather than implausible options like "From work to the gym." Similarly, for the Action Sequence task, we include only two answer candidates corresponding to the actions that actually occurred in the video. More details for the remaining tasks can be found in Appendix A2.
> >
> > ### Strategy 2: Define QA pairs that are not overly informative.
> >
> > Contrary to LLM-based generation, we apply basic templates to mitigate the effect of text-biased QA pairs, addressing Problem 2. Please see Figure 7 in the updated PDF as a summary.
> > - We design QA pairs that are concise and not unnecessarily informative by applying task-specific templates. These templates ensure that the QA pairs lack sufficient information to determine the correct answer purely from text. An example of Unexpected Action is illustrated in Figure 2. QA pairs require the same level of understanding for the model to identify what is amusing in the video but without providing additional textual information. Unlike MVBench, the model cannot simply select the only plausible option containing a dog. We use the same candidate sets across tasks like Action Count, Object Count, Object Shuffle, Action Localization, Unexpected Action, and Moving Direction to ensure balanced datasets with an equal distribution of correct answers, keeping visual complexity while reducing textual bias. Appendix Table 3 provides an overview of all tasks, demonstrating that the QA templates are carefully crafted without unnecessary textual information.
> > - Solving the overreliance on world knowledge requires providing questions and candidates that contain only the necessary information, specifically removing factual information that the LLM can exploit. We remove tasks such as Episodic Reasoning, that are based on QA pairs about TV shows or movies.
> >
> > Thank you again for your feedback. We hope that our answer addresses your concerns and that you will consider raising your score.
> >
> > [1] Kexin Yi, Chuang Gan, Yunzhu Li, Pushmeet Kohli, Jiajun Wu, Antonio Torralba, Joshua B. Tenenbaum: CLEVRER: Collision Events for Video Representation and Reasoning. ICLR 2020
> >
> > [2] Gupta, Adrià Recasens, Larisa Markeeva, Dylan Banarse, Skanda Koppula, Joseph Heyward, Mateusz Malinowski, Yi Yang, Carl Doersch, Tatiana Matejovicova, Yury Sulsky, Antoine Miech, Alexandre Fréchette, Hanna Klimczak, Raphael Koster, Junlin Zhang, Stephanie Winkler, Yusuf Aytar, Simon Osindero, Dima Damen, Andrew Zisserman, João Carreira: Perception Test: A Diagnostic Benchmark for Multimodal Video Models. NeurIPS 2023

---

> > > ### Author Response · Authors · 2024-11-25
> > > **Feedback on Rebuttal**
> > >
> > > Dear Reviewer QEG7,
> > >
> > > Thank you again for the time and effort spent on your thorough review of our paper. Since the author-reviewer discussion deadline is fast approaching, we kindly ask for feedback on our responses. We would be happy to discuss more if there are still some open questions.
> > >
> > > Best Regards,
> > >
> > > Authors

---

### Official Review · Reviewer_A5py · 2024-11-02

**Soundness:** 2
**Presentation:** 2
**Contribution:** 3
**Rating:** 5
**Confidence:** 4

**Summary:**

The paper investigates three issues of MVBench: 1) independence of video or video motion, 2) bias in the generated question-answer pairs, and 3) heavy reliance on world knowledge in questions. A significant part of the paper was written to prove and showcase these problems in the MVBench. A new benchmark called TVBench is proposed to mitigate these issues by redesigning the questions and available choices. The new benchmark attempts to prove that with no visual input or just image input, the models will perform like random guesses. Some strong video models also perform so even with full video inputs. The experiment also presents the results of inputting video frames in reverse order or shuffled order to prove that the benchmark questions requires understanding on the true video motion to be answered correctly.

**Strengths:**

- The paper addresses some critical issues with existing video understanding benchmarks, which is that the correct answers to many questions do not rely on information from the video or video motion. Thus, proposing a new benchmark to resolve these issues are well-motivated.
- The paper explains in detailed examples and some ablation studies to prove that these problems exist widely in MVBench.
- Based solely on the reverse & shuffle order experiment results in Table 4, it seems that TVBench indeed improves some questions' reliance on video inputs and the motion contained in those videos.

**Weaknesses:**

While I appreciate the authors' great efforts to prove their statements of the issues, I am confused by many details after reading through the paper and am not fully convinced by the quality of the new benchmark.

- Some of the results in the figures and tables, or the way they are presented, can be confusing. In Fig.1 left, the trend seems linear after the models achieve a certain level of performance (>50) on MVBench. In Fig. 1, right, MVBench shows a performance drop in VIdeoChat2 when the video is reversed. How do these results support the claim that MVBench does not measure temporal understanding? What is Table 1 trying to prove? I cannot compare the results of GPT-4o + image inputs with Gemini 1.5 Pro + video input to the conclusion that a single image is sufficient. You should at least fix other variables and leave the input as the one changing to prove that. Besides, even though the results are close, did you prove that the questions answered correctly are the same ones? It's a similar issue in Table 2 that I cannot understand how text-only rows could be compared to video input rows since they are using different models.

- Since it's a benchmark, it should attempt to document the performance of as many models as possible. A lot of video models are missing, such as Video-LLaVA, mPLUG-Owl, PandaGPT, ImageBind, Video-LLaMa and etc. In addition, the GPT-4 series can accept multiple images, which is essentially the same as video models with video inputs -- they all need to sample a certain number of frames as multiple image inputs. You can also concatenate multiple frames into one image and feed into the GPT-4 series. It doesn't make sense to me to only benchmark GPT-4o with a single frame input.

- Writing is a big issue in this paper. So many details make it hard to understand the paper without being confused.
  - In Fig. 1, what is the unit of the axes?
  - Table 1 is presented but never referred to in the text. If I understand correctly, some "Tab. 2" should refer to Table 1 instead. Please also choose between "Tab 2" and "Tab. 2" so that searching is convenient.
  - I think the paper shows an excessive amount of bad examples from MVBench, which makes some of these figures unnecessary. While it's good to identify and prove the existence of these problems, more efforts should be spent convincing the readers that the "proposed" benchmark is high-quality and indeed resolves these issues.
  - I understand that Sec. 4 is trying to show that open-ended qa and evaluation are not reliable, but how does that matter with the main point of this paper? Multiple-choice-based QA and open-ended QA are different settings used in different benchmarks or evaluations. It doesn't convince me that TVBench is high quality by showing the weaknesses of open-ended QA -- they are different settings.
  - In line 430, *following the model provided in Tab. 5 for each task*, what is *model* in Table 5? There is no *model* column in Table 5 and the appendix is too short to provide enough context. How many templates are you using? Are the templates in Table 5 showing all you are using? How did you collect these templates? If you have hired annotators, how did you ensure the quality of these templates? These are all important details to be included in the paper to convince readers about the quality of TVBench.
  - This is a minor point, but I don't favor using statistics of **Huggingface downloads** as some sort of evidence in the introduction (lines 37-38). Regardless of whether you are trying to use the number to support MVBench or question its reliability, it's better to appreciate **scientific merits** instead of **popularity metrics** in academic writing.

**Questions:**

Please see the weaknesses.

---

> ### Author Response · Authors · 2024-11-21
> **Rebuttal (1/2)**
>
> Thank you for the time and effort spent in reviewing our paper. In the following, we reply to all the concerns raised in the review.
>
> **W1 – Experiment consistency:** Thank you for addressing this. Below, we respond to your concerns in detail:
>
> ### Model consistency:
> Tables 1 and 2 examine the spatial and temporal biases of MVBench. We have updated these tables in response to your feedback to include the same models across all settings, enabling direct comparisons.
>
> *Table 1: Examining the spatial bias of MVBench using the same models*
>
> || **Input**   | **Fine-grained Action** | **Scene Transition** | **Fine-grained Pose** | **Episodic Reasoning** | **Average** |
> |----------------|---------|---------------------|------------------|-------------------|--------------------|---------|
> | Random   | --  | 25.0| 25.0 | 25.0  | 20.0   | 23.8|
> | |||||
> | Gemini 1.5 Pro |   | 47.0| 78.0 | 46.5  | 56.5   | 57.0|
> | GPT-4o   | image| 49.0| 84.0 | 53.0  | 65.0   | 62.8|
> | Tarsier 34B|   | 48.5| 67.0 | 22.5  | 46.0   | 46.0|
> | |||||
> | Gemini 1.5 Pro |   | 50.0| 93.3 | 58.5  | 66.8   | 67.2|
> | GPT-4o   |video | 51.0| 83.5 | 65.5  | 63.0   | 65.8|
> | Tarsier 34B|   | 48.5| 89.5 | 64.5  | 54.5   | 64.3|
> | |||||
> | Gemini 1.5 Pro | | 49.5| 90.0 | 54.5  | 63.0   | 64.3|
> | GPT-4o   | video shuffle | 52.0  | 84.5 | 69.0  | 64.5   | 67.5|
> | Tarsier 34B|   | 51.0| 89.0 | 56.5  | 51.5   | 62.0|
>
> With the updated table, the message becomes clearer. Models that receive only a random frame as input demonstrate strong performance across all four tasks, surpassing the random baseline. Notably, GPT-4o achieves the highest average performance of 62.8% across these tasks, nearly matching its video-based performance of 65.8%. Overall, GPT-4o gets an average accuracy of 47.8% across all 20 MVBench tasks, which is 20.5% higher than the random baseline of 27.3% (see Table 4). This suggests that a significant portion of the benchmark is influenced by spatial bias. Additionally, shuffling the videos has minimal impact on the performance of all video-language models, with an average difference of only 2.3%, indicating that frame order is not crucial for solving these tasks. This problem goes beyond the four tasks analyzed here. As shown in Table 4, Gemini 1.5 Pro and Tarsier achieve average accuracies of 60.5% and 67.6% across all 20 MVBench tasks, respectively. Shuffling video frames results in performance drops of merely 3.8% and 6.4%, respectively, highlighting that spatial bias affects not only the tasks discussed in this table but the entire dataset.
> Additionally, we verify the agreement between the correct responses of Tarsier 34B across modalities: 91.0% between image and video inputs, and 93.9% between video and shuffled video. This confirms that current models heavily rely on spatial biases to solve MVBench.
>
> *Table 2: Examining the textual bias of MVBench using the same models*
>
> || **Input**| **Action Count** | **Unexpected Action** | **Action Antonym** | **Episodic Reasoning** | **Average** |
> |----------------|------------|--------------|-------------------|----------------|--------------------|---------|
> | Random   | --   | 33.3   | 25.0  | 33.3 | 20.0   | 27.9|
> | |||||
> | Gemini 1.5 Pro | | 49.0   | 68.0  | 85.5 | 49.0   | 62.3|
> | GPT-4o   |  text-only  | 44.0   | 69.5  | 57.5 | 51.5   | 55.6|
> | Tarsier 34B|  | 37.0   | 39.5  | 66.0 | 44.0   | 46.6|
> | |||||
> | Gemini 1.5 Pro |  | 41.2   | 82.4  | 64.5 | 66.8   | 63.7|
> | GPT-4o   | video | 43.5   | 75.5  | 72.5 | 63.0   | 63.6|
> | Tarsier 34B|  | 46.5   | 72.0  | 97.0 | 54.5   | 67.4|
>
> With the updated table, our findings show that models based on text-only can effectively eliminate incompatible candidates, significantly outperforming the random baseline. Notably, models using only text achieves results comparable to video-language models across these four tasks. For example, Gemini 1.5 Pro attains an average performance of 62.3% with text-only input, versus 63.7% when using videos. This trend extends beyond the four tasks, as Gemini 1.5 Pro reaches an average performance of 38.2% across all 20 tasks, which is 10.9% higher than the random chance baseline of 27.3%. We have identified three key sources of this textual bias in the paper.
> Additionally, we verify the agreement between the correct responses of Tarsier 34B across modalities: 85.3% between text and video inputs. This confirms that current models like Tarsier heavily rely on textual biases to solve MVBench.
>
> Furthermore, we have expanded Table 4 to include Gemini 1.5 Pro and Tarsier 34B across all settings (text-only, image, video shuffle, video reverse, and video) for both MVBench and TVBench. Additionally, we examined the spatial and textual biases on another multiple-choice QA benchmark, Next-QA, confirming the same biases observed in MVBench. For more details, see Appendix A.3.
>
>
> --- continued in next comment ---

---

> ### Author Response · Authors · 2024-11-21
> **Rebuttal (2/2)**
>
> ### Clarification of Figure 1
> Figure 1 presents a summary of our paper’s main results. We have updated this figure by incorporating additional models (see W2) and clarified the right side by showcasing only the top-performing video model, Tarsier 34B, for comparisons between MVBench and TVBench under video, shuffle, and reverse settings.
>
> *Right side:* Examines how performance changes when videos are shuffled or reversed on MVBench versus TVBench. On MVBench, altering the frame order results in only minor performance variations, indicating that frame order is not critical for solving MVBench tasks. In contrast, TVBench performance significantly drops when videos are shuffled and drop even further below random chance when videos are reversed. This demonstrates that TVBench requires strong temporal understanding to be solved.
>
> *Left side:* TVBench serves as a strong temporal benchmark, with only a few frontier models surpassing the random chance baseline. In contrast, MVBench exhibits a continuous performance progression across all models. Notably, models that rely solely on a single random frame achieve nearly 50% accuracy, while those based only text (question) reach approximately 40%. These findings raise concerns about what MVBench truly measures, as strong performance can be achieved without genuine temporal understanding.
>
> **W2 – Adding More Video Models to the Benchmark:** We agree with the reviewer that TVBench can benefit from incorporating additional models. Therefore, we have included the suggested models such as GPT-4o with videos, VideoLLaVA, mPLUG-Owl3, VideoLLaMA2 7B, VideoLLaMA2.1 7B, VideoLLaMA2 72B, Qwen2-VL 7B, and Qwen2-VL 72B with more models, e.g. PandaGPT and LLaVA-Next, in the coming days. To ensure the quality of our benchmark, we have also included a human baseline. For more details, see Appendix A.2.2.
> | **Model**  | **MVBench (%)** | **TVBench (%)** |
> |----------------------|-------------------------|-------------------------|
> |Random | 27.3 | 33.3|
> | VideoLLaVA | 42.5  | 33.8  |
> | VideoChat2 | 51.0  | 33.0  |
> | ST-LLM   | 54.9  | 35.3  |
> | GPT-4o   | 49.1  | 39.1  |
> | PLLaVA-7B  | 46.6  | 34.2  |
> | PLLaVA-13B | 50.1  | 35.5  |
> | PLLaVA-34B | 58.1  | 41.9  |
> | mPLUG-Owl3 | 54.5  | 41.4  |
> | VideoLLaMA2 7B | 54.6  | 41.0  |
> | VideoLLaMA2.1 7B | 57.3  | 41.4  |
> | VideoLLaMA2 72B  | 62.0  | 47.5  |
> | VideoGPT+  | 58.7  | 41.5  |
> | Gemini 1.5 Pro | 60.5  | 46.5  |
> | Qwen2-VL 7B| 67.0  | 43.6  |
> | Qwen2-VL 72B   | 73.6  | 52.5  |
> | Tarsier-7B | 62.6  | 45.8  |
> | Tarsier 34B| 67.6  | 53.8  |
> |Human Baseline |-- | 94.8 |
>
> **W3 – Presentation:**
> Thank you for the detailed feedback on our paper. In response to your suggestions, we have made the following updates to our revised PDF:
> - Minor Adjustments: Added the axis to Fig. 1 and corrected the reference to Table 1 to improve clarity and accuracy.
> - Reducing MVBench examples: We removed one of the three examples of MVBench in Fig. 2
> - Human Baseline: Included a human baseline to verify the quality and solvability of our benchmark.
> - Benchmark Examples: Added examples of TVBench for all tasks in Appendix A3 to give more insights.
> - Model Consistency: Ensured consistency by including the same models across all modalities in Tables 1, 2, and 4, facilitating direct comparisons.
> - Benchmark Creation Details: Clarified the strategies for creating the benchmark in Section 5.1, including the addition of Figure 7 to visually represent our process.
> - Extended MCQA Analysis: Expanded our analysis beyond MVBench by also evaluating NextQA, as detailed in Appendix A.3, exhibiting the same problems of spatial and temporal biases.
>
> *Importance of Section 4 (Open-Ended VQA):* Currently, video-language models are evaluated using two methods: multiple-choice and open-ended VQA. After examining the widely used MVBench benchmark for multiple-choice tasks—and extending our analysis to NextQA in Appendix A.3—we found that spatial and textual biases also exist in open-ended VQA. Additionally, relying on closed-source proprietary LLMs for evaluation introduces unreliability. This discovery motivated us to focus on designing a multiple-choice VQA benchmark rather than a new open-ended VQA. As appreciated by Reviewer TPmi, we believe that highlighting these issues in open-ended VQA is beneficial for the community.
>
>
> Thank you again for reviewing our paper and providing constructive feedback. We hope that our responses have addressed your concerns and that you will consider increasing your score.

---

> > ### Author Response · Authors · 2024-11-25
> > **Feedback on Rebuttal**
> >
> > Dear Reviewer A5py,
> >
> > Thank you again for the time and effort spent on your thorough review of our paper. Since the author-reviewer discussion deadline is fast approaching, we kindly ask for feedback on our responses. We would be happy to discuss more if there are still some open questions.
> >
> > Best Regards,
> > Authors

---

> > > ### Author Response · Authors · 2024-11-27
> > > **Extended Rebuttal**
> > >
> > > Dear Reviewer A5py,
> > >
> > > As per the reviewer’s suggestion and in continuation of our previous response, we have expanded our analysis by including additional video-language models. The updated list now features the latest models, including LLaVA-Video 7B, LLaVA-Video 72B, Aria, and IXC-2.5-7B. Below is the revised table, presenting the performance of all models on MVBench and TVBench:
> > >
> > > | **Model**            | **MVBench (%)** | **TVBench (%)** |
> > > |----------------------|-----------------|-----------------|
> > > | Random | 27.3           | 33.3           |
> > > | VideoLLaVA | 42.5           | 33.8           |
> > > | VideoChat2 | 51.0           | 33.0           |
> > > | ST-LLM | 54.9           | 35.3           |
> > > | GPT-4o | 49.1           | 39.1           |
> > > | PLLaVA-7B | 46.6           | 34.2           |
> > > | PLLaVA-13B | 50.1           | 35.5           |
> > > | PLLaVA-34B | 58.1           | 41.9           |
> > > | mPLUG-Owl3 | 54.5           | 41.4           |
> > > | VideoLLaMA2 7B | 54.6           | 41.0           |
> > > | VideoLLaMA2.1 7B | 57.3           | 41.4           |
> > > | VideoLLaMA2 72B | 62.0           | 47.5           |
> > > | VideoGPT+            | 58.7           | 41.5           |
> > > | Gemini 1.5 Pro | 60.5           | 46.5           |
> > > | Qwen2-VL 7B | 67.0           | 43.6           |
> > > | Qwen2-VL 72B | 73.6           | 52.5           |
> > > | LLaVA-Video 7B | 58.6           | 45.2           |
> > > | LLaVA-Video 72B | 64.1           | 49.6           |
> > > | Aria | 69.7           | 50.5           |
> > > | IXC-2.5-7B | 69.1           | 50.5           |
> > > | Tarsier-7B | 62.6           | 45.8           |
> > > | Tarsier 34B | 67.6           | 53.8           |
> > > | Human Baseline | -- | 94.8 |
> > >
> > > We kindly request any feedback on our responses and would be happy to address any remaining questions or concerns.
> > > Thank you for your time and consideration.
> > >
> > > Best regard, Authors

---

### Official Review · Reviewer_sqwF · 2024-11-04

**Soundness:** 3
**Presentation:** 3
**Contribution:** 2
**Rating:** 3
**Confidence:** 2

**Summary:**

The paper introduces TVBench, a new video-language benchmark that addresses critical flaws in existing benchmarks like MVBench. The authors identified three problems with current benchmarks such as:
- Single frames are enough
- Question text reveals answers.
- Common knowledge beats video.

The authors demonstrated those problems by showing that both text-only language models and single-frame vision models perform well on existing benchmarks. In contrast, when it comes to TVBench, most state-of-the-art video-language models perform close to random chance.

The benchmark consists of 10 temporal tasks across 2,654 question-answer pairs, ensuring models must understand the sequence and timing of video events to succeed. The authors validated their benchmark by showing that shuffling or reversing video frames significantly impacts performance, unlike previous benchmarks.

**Strengths:**

- MVBench Analysis: Thorough and systematic identification of MVBench limitations with clear evidence.
- Validation Methods: Creative use of video shuffling/reversal to verify temporal understanding requirements.
- Benchmark difficulty: Evaluation showing most current models fail at true temporal reasoning.

**Weaknesses:**

- Problem analysis: even though the authors did identify the problems with MVBench, the analysis of other benchmarks is quite limited. The paper states “We conduct a comprehensive analysis of widely used video question-answering benchmarks” while focusing only on MVBench. Several datasets in the relative section can be analyzed similarly and it is still uncertain if all of those datasets also have those problems
- Small amount of dataset examples: there are 10 different tasks within the dataset, yet the paper shows only one example from the whole dataset.
- Task design: Authors state: “Questions should not be answerable using spatial details from a single random frame or multiple frames e.g. after shuffling them.” However, even the only given example from TVBench about scenes in the movie can be solved using two frames. Additionally, tasks in TVBench like Scene transition, Action Antonym, and Moving Direction can be solved with only two frames instead of one. Image LLM evaluation with more frames would be important.
- Benchmark creation details: There are very few details on how the dataset was collected and annotated. In general, given details about the dataset creation are very vague. For example: "Instead of including random, easy negative candidates, we define hard candidates that cannot be discarded without temporal information". How do you generate the hard negative examples?

**Questions:**

- How the dataset was annotated, how exactly did you come up with the wrong answers?
- Show more examples from the dataset. Several examples from each of the tasks. The examples can be illustrated similarly to how you did it in Figure 5
- Are those MVBench problems shown in other benchmarks?

**Details Of Ethics Concerns:**

no ethics concerns

---

> ### Author Response · Authors · 2024-11-21
> **Rebuttal (1/2)**
>
> Thank you for the time and effort spent in reviewing our paper. In the following, we respond to your comments and how they have further strengthened our submission.
>
> **W1 & Q3 - Problems also present in other benchmarks?**: In Section 3, we focused on MVBench as it is widely used for multiple-choice video question-answering. In Section 4, we broadened our analysis to three other benchmarks (MSVD-QA, MSRVTT-QA, ActivityNet-QA) for open-ended video QA. We found these benchmarks exhibit similar issues as MVBench (e.g. video shuffle and video perform almost similarly well on ActivityNet-QA (see Table 3). Moreover, we show that these can be unreliable due to reliance on closed API LLMs for evaluation. Additionally, we have now analyzed the NextQA [1] multiple-choice benchmark (see Appendix A.3), which shows the same pattern of strong textual and spatial biases:
> | Model  | Input| NextQA (%) |
> |-------|----------------|--------|
> |Random  | -- | 20.0   |
> |Tarsier 34B  | text-only  | 47.6   |
> |Tarsier 34B | image| 71.3   |
> |Tarsier 34B | video shuffle  | 78.5   |
> |Tarsier 34B | video reverse  | 77.6   |
> |Tarsier 34B | video| 79.0   |
>
> As shown, Tarsier 34B achieves an accuracy of 71.3% using only a single image, nearly matching its 79.0% accuracy with full video input, indicating a strong spatial bias, similar to what we observed with MVBench. Also, shuffling or reversing video frames does not impact performance, similarly to MVBench, demonstrating temporal frame consistency is not needed.
>
> **W2 & Q2 – TVBench Examples:** Thanks for pointing this out. In Appendix A.4, we add Fig. 7-13 providing 45 examples from our benchmark. In addition, we also provide more examples of the spatial and textual bias in MVBench in Fig. 14-25.
>
> **W3 – Task Design:** Thank you for highlighting this point. It is possible that selecting the correct two frames is sufficient to address the Scene Transition task. However, the model must accurately identify and interpret these frames, including their order, making it a temporal challenge. To verify this, we report the performance of the leading method, Tarsier 34B, on TVBench as follows:
>
> | Setting   | TVBench (%) |
> |-|---------|
> | Random| 33.3%|
> | Single Random Frame | 35.0%|
> | Two Random Frames   | 36.8%|
> | Video   | 53.8%|
>
> We find that TVBench cannot be effectively addressed using two random frames. Although scene transition might appear to be a straightforward task, many recent models struggle with it even when given the entire video, performing nearly at chance levels—for instance, GPT-4o achieves 39.1%, which is only 8.8% better than random guessing. Additionally, we evaluate the average performance on MVBench using two random frames with Tarsier 34B:
>
> | Setting   | MVBench (%) |
> |---|--------------|
> | Random | 27.3%  |
> | Single Random Frame | 45.1%  |
> | Two Random Frames   | 56.5%  |
> | Video   | 67.6%  |
>
> These results on MVBench indicate that using two random frames significantly improves performance by 11.4% compared to a single frame, approaching the video-level performance of 67.6%. In contrast, on TVBench, the performance with two random frames remains close to random chance. This further underscores the necessity of a challenging temporal benchmark like TVBench.
>
> --- continued in next comment ---

---

> > ### Author Response · Authors · 2024-11-21
> > **Rebuttal (2/2)**
> >
> > **W4 & Q1 – Benchmark Creation Details:** Thank you for highlighting this. Below, we outline the two primary strategies used to create our benchmark, which have now been incorporated into Section 5.1 of the paper. We base our QA templates on the original dataset annotations for each task and do not require annotators. Detailed information for each task, including templates, questions and answers, and video statistics, is provided in Appendix A.2.
> >
> > ### Strategy 1: Define Temporally Hard Answer Candidates.
> > To address Problem 1, the temporal constraints in the question must be essential for determining the correct answer. This involves designing time-sensitive questions and selecting temporally challenging answer candidates.
> > - We select 10 temporally challenging tasks that require: Repetition counting (Action Count), Properties of moving objects (Object Shuffle, Object Count, Moving Direction), Temporal localization (Action Localization, Unexpected Action), Sequential ordering (Action Sequence, Scene Transition, Egocentric Sequence), distinguishing between similar actions (Action Antonyms).
> > - We define hard-answer candidates based on the original annotations to ensure realism and relevance, rather than relying on LLM-generated candidates that are often random and easily disregarded, as seen in MVBench. For example, in the Scene Transition task (see Figure 6), we design a QA template that provides candidates based on the two scenes occurring in the videos for this task, rather than implausible options like "From work to the gym." Similarly, for the Action Sequence task, we include only two answer candidates corresponding to the actions that actually occurred in the video. More details for the remaining tasks can be found in Appendix A2.
> >
> > ###  Strategy 2: Define QA pairs that are not overly informative.
> > Contrary to LLM-based generation, we apply basic templates to mitigate the effect of text-biased QA pairs, addressing Problem 2. Please see Figure 7 in the updated PDF as a summary.
> > - We design QA pairs that are concise and not unnecessarily informative by applying task-specific templates. These templates ensure that the QA pairs lack sufficient information to determine the correct answer purely from text. An example of Unexpected Action is illustrated in Figure 2. QA pairs require the same level of understanding for the model to identify what is amusing in the video but without providing additional textual information. Unlike MVBench, the model cannot simply select the only plausible option containing a dog. We use the same candidate sets across tasks like Action Count, Object Count, Object Shuffle, Action Localization, Unexpected Action, and Moving Direction to ensure balanced datasets with an equal distribution of correct answers, keeping visual complexity while reducing textual bias. Appendix Table 3 provides an overview of all tasks, demonstrating that the QA templates are carefully crafted without unnecessary textual information.
> > - Solving the overreliance on world knowledge requires providing questions and candidates that contain only the necessary information, specifically removing factual information that the LLM can exploit. We remove tasks such as Episodic Reasoning, that are based on QA pairs about TV shows or movies.
> >
> > Thank you for your effort in reviewing our paper and the feedback provided which we believe has strengthened our work. We are happy to further discuss and if these points are answered we ask that you consider increasing your score to reflect the revisions and clarifications.
> >
> > [1] NExT-QA: Next Phase of Question-Answering to Explaining Temporal Actions. Xiao, J., Shang, X., Yao, A., & Chua, T.-S. CVPR 2021.

---

> > > ### Author Response · Authors · 2024-11-25
> > > **Feedback on rebuttal**
> > >
> > > Dear Reviewer sqwF,
> > >
> > > Thank you again for the time and effort spent on your thorough review of our paper. Since the author-reviewer discussion deadline is fast approaching, we kindly ask for feedback on our responses. We would be happy to discuss more if there are still some open questions.
> > >
> > > Best Regards,
> > > Authors

---

### Official Review · Reviewer_TPmi · 2024-11-04

**Soundness:** 3
**Presentation:** 4
**Contribution:** 3
**Rating:** 6
**Confidence:** 4

**Summary:**

This paper introduces TVBench, a new benchmark for testing video understanding capability of multimodal models. Flaws in widely used existing benchmark (MVBench) are demonstrated, namely, spatial biases, textual biases and reliance on world knowledge. In addition, it's also shown that open-ended benchmarks can contain similar biases. TVBench is constructed from pre-defined templates in order to mitigate these biases and test temporal reasoning capabilities. Supporting experimental results demonstrate that state-of-the-art models struggle on this benchmark. Similarly, text-only or image-text foundation models struggle to beat random chance signifying the difficulty of this benchmark compared with existing benchmark.

**Strengths:**

The paper is well-written and easy to understand. It tackles an important area of video understanding, i.e. the lack of strong benchmarks that test temporal reasoning in videos. The presentation clearly analyzes drawbacks of existing benchmarks and proposes a new benchmark.
- The QA pairs don't use LLMs in the loop, and thus can avoid many hallucination related issues.
- The performance of SOTA models is very low (Table 4). This indicates the benchmark is indeed difficult.
- Clear contrast with MVBench is demonstrated, especially using text-only and image-only models. This justifies most of the claims in the paper.
- A significant, and often overlooked issue in open-ended evaluations is pointed out in Section 4. Using closed-source proprietary models whose back-ends may change arbitrarily to score open-ended responses and track our progress on video understanding can be misleading.

**Weaknesses:**

The main weakness of this work is around experimentation.
- Human baseline performance is not presented. This is important to judge the quality of the benchmark and the presented results.
- Different models are used in Table 2 to make the claim that MVBench has textual bias. Ideally, the same model (ideally the best model) needs to be presented with text-only and video as inputs to justify the claim.
- Similarly, in Table 4, different models are used to compare different biases (text, image, video) of the model.

Further Limitations:
- Using standard template QA pairs may limit the range of video understanding being assessed.
- In Figure 2 and the associated text in the paper, it's presented as if detecting the absence of something is an easy task. However, by definition, one must watch the entire video to make sure what we're detecting is indeed absent.

**Questions:**

- Can we use the same model and ablate text-only, image, video, shuffle, reverse, etc. in Table 4? Ideally Gemini 1.5 pro as it performs the best on this benchmark?
- MVBench is presented as not a great benchmark. However, its performance is also not saturated (best model achieves 67.7 in Table 4). Do the remaining QA pairs satisfy the criteria set in the paper? What is the size of the data? Can we remove the bad examples from MVBench and get a bigger and better dataset than TVBench?

EDIT: Updated score based on the rebuttal.

---

> ### Author Response · Authors · 2024-11-21
> **Rebuttal (1/3)**
>
> Thank you for the time and effort spent in reviewing our paper. In the following, we reply to all the concerns raised in the review.
>
> **W1 - Human baseline:**
> Thank you for pointing it out. We used 14 annotators to establish a human baseline for TVBench. Each annotator labeled around 30 videos. The overall accuracy of the human baseline for our benchmark is 95%. In our updated Appendix we have also computed the statistics for the mean error for this task: 4.5% using finite population correction, thus verifying the quality and solvability of our benchmark.
>
> **W2 & W3 & Q1 Model consistency:** Thank you for your suggestion. We updated Tables 1, 2, and 4 using the same models (GPT-4o, Gemini 1.5 Pro, and Tarsier 34B) across all modalities enabling a direct comparison.
>
> For convenience, we attach the updated tables here:
>
> *Table 1: Examining the spatial bias of MVBench using the same models*
>
> || **Input**   | **Fine-grained Action** | **Scene Transition** | **Fine-grained Pose** | **Episodic Reasoning** | **Average** |
> |----------------|---------|---------------------|------------------|-------------------|--------------------|---------|
> | Random   | --  | 25.0| 25.0 | 25.0  | 20.0   | 23.8|
> | |||||
> | Gemini 1.5 Pro |image| 47.0| 78.0 | 46.5  | 56.5   | 57.0|
> | GPT-4o   | image| 49.0| 84.0 | 53.0  | 65.0   | 62.8|
> | Tarsier 34B|  image | 48.5| 67.0 | 22.5  | 46.0   | 46.0|
> | |||||
> | Gemini 1.5 Pro | video| 50.0| 93.3 | 58.5  | 66.8   | 67.2|
> | GPT-4o   |video | 51.0| 83.5 | 65.5  | 63.0 | 65.8|
> | Tarsier 34B| video  | 48.5| 89.5 | 64.5  | 54.5| 64.3|
> | |||||
> | Gemini 1.5 Pro | video shuffle| 49.5| 90.0 | 54.5  | 63.0   | 64.3|
> | GPT-4o| video shuffle | 52.0| 84.5 | 69.0  | 64.5   | 67.5|
> | Tarsier 34B| video shuffle| 51.0| 89.0 | 56.5| 51.5 | 62.0|
>
> *Table 2: Examining the textual bias of MVBench using the same models*
>
> || **Input**| **Action Count** | **Unexpected Action** | **Action Antonym** | **Episodic Reasoning** | **Average** |
> |-|-|-|-|-|-|-|
> | Random| -- | 33.3| 25.0  | 33.3 | 20.0| 27.9|
> | |||||
> | Gemini 1.5 Pro | text-only | 49.0   | 68.0  | 85.5 | 49.0| 62.3|
> | GPT-4o   |  text-only  | 44.0   | 69.5  | 57.5 | 51.5| 55.6|
> | Tarsier 34B|text-only| 37.0   | 39.5  | 66.0 | 44.0| 46.6|
> | |||||
> | Gemini 1.5 Pro | video | 41.2   | 82.4  | 64.5 | 66.8   | 63.7|
> | GPT-4o| video | 43.5   | 75.5  | 72.5 | 63.0   | 63.6|
> | Tarsier 34B| video | 46.5   | 72.0  | 97.0 | 54.5   | 67.4|
>
> *Table 4: Benchmark overview with the same model.*
> For a full table with all models and all TVBench tasks please see the updated PDF. We omitted GPT-4o for shuffle and reverse for cost reduction.
>
> | **Model**  | **Input**| **MVBench Average (%)** | **TVBench Average (%)** |
> |--|-|---|-|
> | Random| –| 27.3 |33.3|
> |  ||   |   |
> | GPT-4o |Text-only | 34.8| 33.8|
> | Gemini 1.5 Pro   | Text-only  | 38.2| 33.6|
> | Tarsier 34B  |Text-only| 35.7| 34.4|
> |  ||   |   |
> | GPT-4o |  Image  | 47.8| 35.8|
> | Gemini 1.5 Pro   | Image| 48.5| 36.3|
> | Tarsier 34B  |  Image  | 45.1| 35.0|
> |  || ||
> | Gemini 1.5 Pro   | Video Reverse  | 53.1| 27.0|
> | Tarsier 34B  |  Video Reverse  | 67.7| 27.2|
> |  ||   |   |
> | Gemini 1.5 Pro   | Video Shuffle  | 56.8| 36.1|
> | Tarsier 34B  | Video Shuffle | 61.2| 38.0|
> |  ||   |   |
> | GPT-4o |Video | 49.1| 39.1|
> | Gemini 1.5 Pro   | Video| 60.5| 46.5|
> | Tarsier 34B |Video| 67.6| 53.8|
>
> ### With that, our message becomes even clearer:
> *Spatial Bias:*
>
> - On MVBench, models that receive only a random frame as input demonstrate strong performance across all four tasks, surpassing the random baseline (Table 1). Notably, GPT-4o achieves the highest average performance of 62.8% across these tasks, nearly matching its video-based performance of 65.8%. This issue extends beyond these four tasks, as GPT-4o attains an average accuracy of 47.8% across all 20 MVBench tasks, which is 20.5% higher than the random baseline of 27.3% (Table 4). This indicates that a significant portion of the benchmark is influenced by spatial bias. Similarly, for Gemini 1.5 Pro and Tarsier.
>     - In contrast, on TVBench, GPT-4o with a random frame performs close to random chance, achieving only a 1.7% improvement over the baseline. This verifies that TVBench is a strong new benchmark that cannot be solved with a single random frame.
> - When videos in MVBench are shuffled or reversed, the performance of top models like Tarsier-34B and Gemini 1.5 Pro remains largely unchanged. Specifically, Tarsier-34B maintains performance levels of 61.2% and 67.7%, which are 33.9% and 40.4% above the random baseline, respectively (Table 4). This consistency indicates that the order of frames does not matter for MVBench.
>     -In contrast, shuffling and reversing videos in TVBench result in substantial performance declines of 38.0% and 27.2% which is only 4.7% better and even 6.1% worse than the random baseline respectively. These results demonstrate that frame order is crucial for TVBench, as reversing the sequence of frames leads to incorrect answers.
>
> --- continued ---

---

> ### Author Response · Authors · 2024-11-21
> **Rebuttal (2/3)**
>
> *Textual Bias:*
>
> - MVBench exhibits a strong textual bias as models using only text (question) achieve competitive results compared to those with video input across the four tasks in Table 2. For instance, Gemini 1.5 Pro (text-only) achieves an average performance of 62.3% nearly matching its video-based performance of 63.7%. This issue goes beyond the four tasks, as Gemini Pro 1.5 (text-only) achieves an average performance across all 20 tasks of 38.2%, which is 10.9% higher than the random chance baseline of 27.3%, see Table 4.
>     - On TVBench, models with only text, such as Gemini 1.5 Pro, only improve 0.3% above the random chance baseline, verifying that questions cannot be answered without the corresponding videos.
>
> The new tables can be found in the updated PDF. We further improved our benchmark by evaluating the newest video language models such as Qwen2-VL, Video-LLaMA2, Video-LLaMA2.1, Video-LLaVA, mPLUG-Owl3 on our benchmark.
>
>
> **L1 Standard template QA:**
> In the era of modern LLMs, providing unnecessary context in questions or answer candidates can lead to mis-evaluations. We observed this in MVBench, where LLMs were used to create answer candidates, some of which are non-sensical. Hiring annotators to propose answer candidates is however expensive. Therefore, we decided to adopt standard templates specifically designed for each of the 10 tasks to eliminate any spatial or textual bias. With this approach, we created a temporally challenging benchmark in which the most recent methods achieve only 53.7%, representing an improvement of ~20% over the random chance baseline.
>
>
>
> **Q2 Reminder of MVBench:**
>
> ### MVBench is not yet saturated, why TVBench?
> Thank you for pointing this out. While MVBench is not saturated, it is unclear what it truly measures, as it exhibits both strong spatial and textual biases. For instance, in Fig. 3, example 5, the Episodic Reasoning task requires extensive world knowledge to answer the question about the TV show but lacks any temporal component. Similarly, the Fine-grained Action Recognition task in Fig. 2 only requires image understanding of the bathtub, with no need for temporality. This issue extends beyond individual examples, as image-based models achieve nearly 50% accuracy on MVBench.
> In summary, MVBench assesses various aspects of multi-modal understanding but fails to specifically measure temporality. This limitation motivated us to develop TVBench, explicitly designed to evaluate temporal understanding, as confirmed through experiments such as shuffling, and analyzing image and text accuracies.
> ### Cleaning MVBench for a larger benchmark?
> The problems with MVBench are more structural than simply cleaning bad examples. To address this, we analyzed MVBench and selected tasks suitable for a temporal benchmark—those that are unsaturated and temporally challenging—but required new templates for generating question-answer pairs to mitigate spatial and textual biases. We also sourced videos from original datasets like Perception Test, CLEVRER, and STAR, designing new task-specific templates based on the original annotations. Below are the tasks we retained from MVBench but redesigned:
> - Object Shuffle (OS):This task effectively evaluates temporal understanding. QA pairs are sourced from the Perception Test and supplemented with additional pairs to ensure balanced answer distributions, avoiding correlations that might bias models.
> - Action Count (AC): Similar to OS, this task evaluates temporal reasoning but suffers from imbalanced answers (e.g., one answer appears 45% of the time). We balance the dataset by adding more QA pairs from the original source.
> - Action Localization (AL): QA pairs from MVBench are reused but filtered to remove correlations between question verbs (e.g., "open") and answers (e.g., "At the beginning of the video"). The set is rebalanced to minimize textual bias.
> - Moving Direction (MD): Additional QA pairs are sourced from CLEVRER, excluding stationary object answers to ensure temporal reasoning is required. MVBench's ChatGPT-based QA generation strategy is avoided.
> - Scene Transition (SC): Videos are reused, and QA pairs are rephrased to include only challenging candidates. Options are restricted to scenes that appear in the video, reducing spatial bias.
> - Action Sequence (AS): Videos are sourced from STAR, as in MVBench, but answer options are limited to actions that actually occur in the video to address spatial bias.
> - Unexpected Action (UA): Videos from FunQA are used with a new QA generation strategy. Instead of describing actions, the task requires localizing unexpected actions, avoiding biases from ChatGPT-generated QA pairs.
>
> --continued--

---

> ### Author Response · Authors · 2024-11-21
> **Rebuttal (3/3)**
>
> For a comprehensive temporal benchmark, we introduced the following tasks to expand the benchmark:
> - Object Count (OC): Videos are sourced from CLEVRER, similar to the Moving Count tasks in MVBench. However, unlike MVBench, ignoring the temporal aspects of the question leads to incorrect answers in TVBench. Answers are also balanced to ensure fairness.
> - Action Antonym (AA): While MVBench includes an Action Antonym task, we have completely reformulated it. Videos are sourced from a different dataset (NTU RGB-D instead of PAXION), and temporally opposed candidates (e.g., "sitting down" vs. "standing up") are generated, replacing textually opposed ones, as shown in examples 1 and 2 of Fig. 3.
> - Egocentric Sequence (ES): Videos are sourced from the CSV dataset (not utilized in MVBench), featuring detailed action sequences recorded from a first-person perspective. To evaluate temporal understanding, negative candidates are created by reordering the correct sequence of actions.
>
> **L2 Absence detection:**
> Thank you for pointing this out. Indeed, correctly classifying the presence or absence of an object requires analyzing all frames of a video. However, this can be achieved by treating each frame independently, as this task does not require temporal understanding of the video. State-of-the-art models like Tarsier-7B and Tarsier 34B effectively solve this task, achieving accuracies of 95.0% and 96.5% on the Object Existence (OE) task in MVBench, respectively. Thus, in order to track progress in challenging video understanding cases, we have chosen to not incorporate this task into our benchmark.
>
> Thank you again for your feedback! We hope that our answer addresses your concerns about experimentation and that you will consider raising your score.

---

> > ### Author Response · Authors · 2024-11-25
> > **Feedback on rebuttal**
> >
> > Dear Reviewer TPmi,
> >
> > Thank you again for the time and effort spent on your thorough review of our paper. Since the author-reviewer discussion deadline is fast approaching, we kindly ask for feedback on our responses. We would be happy to discuss more if there are still some open questions.
> >
> > Best Regards,
> > Authors

---

> > > ### Comment · Reviewer_TPmi · 2024-11-29
> > >
> > > I'd like to thank the authors for the detailed rebuttal. Most of my concerns have been addressed, especially (a) human baseline and (b) using the same model consistently across all the ablations. Looking at the updated results, I think the claims on the drawbacks of MVBench have weakened (based on performance difference between image and video models, drop in performance due to shuffling etc.) and I'd like to request the authors to update the text accordingly. Nevertheless, just based on updated Table 4, TVBench looks like a strong benchmark.
> > >
> > > RE Cleaning MVBench?
> > > Since Table 4 already has the set of examples in MVBench that can be answered using text-only, image-only etc., can the authors filter MVBench and provide the number of remaining examples, and the performance on the remaining examples, of say Gemini 1.5 pro? This will clearly demonstrate the marginal impact of this work.

---

> > > > ### Author Response · Authors · 2024-11-29
> > > > **Response**
> > > >
> > > > Thank you for your response and for acknowledging TVBench as a strong benchmark.
> > > >
> > > >
> > > > **Re. MVBench claims weakened:** We respectfully disagree with this. The performance drop across different settings, such as image or shuffled inputs, remains significant. The possibility of any model (e.g., Gemini 1.5 Pro) solving multiple tasks (Table 1) equally well using only a single random frame rather than the full video is very concerning. Even more strikingly, Gemini achieves nearly 50% accuracy on the entire MVBench dataset with just a single random frame as input—21% above the random chance baseline. Moreover, when frames are shuffled, Gemini’s performance drops only slightly from 60.5% to 56.8%, clearly showing that temporal aspects of the videos can be disregarded. These findings demonstrate that MVBench cannot reliably assess the temporal understanding of video-language models. Therefore, it does not meet the criteria for a robust temporal benchmark that our community should adopt.
> > > >
> > > >
> > > > **Re. Cleaning MVBench?:**
> > > > As suggested, we exclude samples that can be correctly answered using text-only, single-image, and shuffled-video inputs for Gemini 1.5 Pro. This leaves only 978 samples (24.7% of MVBench). On this subset, Gemini 1.5 Pro, which has proven to be a strong model in TVBench, achieves a performance of 27.3%, equivalent to random chance. This indicates that all samples correctly answered by Gemini 1.5 Pro are influenced by spatial or textual biases. The remainder fails as a temporal benchmark since performance aligns with random chance, further emphasizing the need for a more robust benchmark like TVBench.
> > > >
> > > >
> > > > Thank you once again for your valuable feedback. We hope our response effectively addresses your remarks, and we remain open to further discussion if needed.

---

> > > > > ### Comment · Reviewer_TPmi · 2024-12-02
> > > > >
> > > > > Thanks for the response!
> > > > >
> > > > > I feel like most benchmarks have both easy and hard examples, and the technology innovations are in tackling the remaining hard examples after the easy ones are solved by existing methods. In this sense, I do not consider that MVBench fails to meet the criteria of a robust temporal benchmark. From your experiments, cleaned-up version of MVBench being close to random chance suggests that the model (Gemini) doesn't yet have a robust temporal understanding capability, and this is the characteristic of a strong benchmark.
> > > > >
> > > > > For this reason, I would suggest to reconsider the claims on MVBench.

---

> ### Author Response · Authors · 2024-12-04
> **Response**
>
> Thank you for your comment.
>
> We redesigned the tasks in TVBench from the ground up to ensure they are inherently temporally challenging, rather than relying on state-of-the-art methods to identify “hard” samples in existing benchmarks. Consequently, some methods perform at random baseline levels, while only those with robust temporal reasoning, like Gemini, outperform this baseline. Additionally, reversing the frame order causes models like Gemini to perform below random levels, indicating that our benchmark requires temporal understanding. This highlights a limitation of the filtered MVBench subset, where performance reflects random chance rather than the method’s (e.g., Gemini’s) capabilities.

---

### Author Response · Authors · 2024-11-21
**Global Response**

We thank the reviewers for dedicating their time and effort to reviewing our paper and for providing their thoughtful feedback. The positive reception of our paper addressing an important problem in video-language evaluation (TPmi, A5py, QEG7) that will have a positive and high impact on the community (QEG7) is highly encouraging. Additionally, we are pleased that the presentation was found to be clear and convincing (QEG7), effectively demonstrating the limitations of existing benchmarks like MVBench (TPmi, sqwF, A5py, QEG7). Our new benchmark TVBench is recognized as both difficult (TPmi, A5py) and temporally challenging (QEG7), revealing that most models fail at true temporal reasoning (sqwF). Lastly, we appreciate the acknowledgment of pointing out a significant and overlooked issue of open-ended evaluation (TPmi).

Below, we address each of the reviewers’ comments individually and look forward to engaging in a constructive discussion during the author-reviewer discussion period.
Thank you once again.

---

### Comment · Area_Chair_La1x · 2024-12-03
**Discussion due soon**

Dear all reviewers,

Our reviewer-author discussion will end soon. For each of you, please check all the files and see if anything you'd like to discuss with authors.

Best, Your AC

---

### Meta-Review · Area_Chair_La1x · 2024-12-16

**Metareview:**

This paper proposes a video model evaluation by identifying three issues with solutions. It received mixed reviews. [TPmi, QEG7] are positive while the other two ([sqwF, A5py]) are negative. The raised issues reside in task configuration and benchmark construction details. In the rebuttal phase, the authors try to address these issues by providing more experiments and specific explanations. Overall, the AC has checked the files, and agrees with [sqwF, A5py] that task setting is a bit of unclear (e.g., frame utilization), and the benchmark detailed explanations need improvement. The authors shall better prepare the current presentation and setting, and welcome for the next venue.

**Additional Comments On Reviewer Discussion:**

[sqwF] raised problem definition, small amount of data, task design unclear, lack of benchmark creation details. The authors responded by showing other benchmarks analysis, adding 45 examples, analyzing two frames configuration, and illustration dataset details. These aspects, overall, fall within the similar design of existing works and do not show the clear contribution w.r.t prior arts. On the other hand, [A5py] points out many writing issues, which are partially addressed by the authors, with concerns remained.

---

### Decision · Program_Chairs · 2025-01-22

Reject